# Holistic Molecular Representation Learning via Multi-view Fragmentation

**Seojin Kim***     *osikjs@kaist.ac.kr*
*Korea Advanced Institute of Science & Technology (KAIST)*

**Jaehyun Nam***     *jaehyun.nam@kaist.ac.kr*
*Korea Advanced Institute of Science & Technology (KAIST)*

**Junsu Kim**     *junsu.kim@kaist.ac.kr*
*Korea Advanced Institute of Science & Technology (KAIST)*

**Hankook Lee**     *hankook.lee@skku.edu*
*Sungkyunkwan University (SKKU)*

**Sungsoo Ahn**     *sungsoo.ahn@postech.ac.kr*
*Pohang University of Science and Technology (POSTECH)*

**Jinwoo Shin**     *jinwoos@kaist.ac.kr*
*Korea Advanced Institute of Science & Technology (KAIST)*

*\* The authors contributed equally.*

**Reviewed on OpenReview:** *https://openreview.net/forum?id=ufDh55J1ML*

## Abstract

Learning chemically meaningful representations from unlabeled molecules plays a vital role in AI-based drug design and discovery. In response to this, several self-supervised learning methods have been developed, focusing either on global (e.g., graph-level) or local (e.g., motif-level) information of molecular graphs. However, it is still unclear which approach is more effective for learning better molecular representations. In this paper, we propose a novel holistic self-supervised molecular representation learning framework that effectively learns both global and local molecular information. Our key idea is to utilize *fragmentation*, which decomposes a molecule into a set of chemically meaningful fragments (e.g., functional groups), to associate a global graph structure to a set of local substructures, thereby preserving chemical properties and learn both information via contrastive learning between them. Additionally, we also consider the 3D geometry of molecules as another view for contrastive learning. We demonstrate that our framework outperforms prior molecular representation learning methods across various molecular property prediction tasks.

## 1 Introduction

Obtaining discriminative representations of molecules is a long-standing problem in chemistry (Morgan, 1965). Such a task is critical for many applications, such as drug discovery (Capecchi et al., 2020) and material design (Gómez-Bombarelli et al., 2018), since it is a fundamental building block for various downstream tasks, e.g., molecular property prediction (Duvenaud et al., 2015) and molecular generation (Mahmood et al., 2021). Over the past decades, researchers have focused on handcrafting the molecular fingerprint representation which encodes the presence of chemically informative substructures, e.g., functional groups, in a molecule (Rogers & Hahn, 2010; Seo et al., 2020).

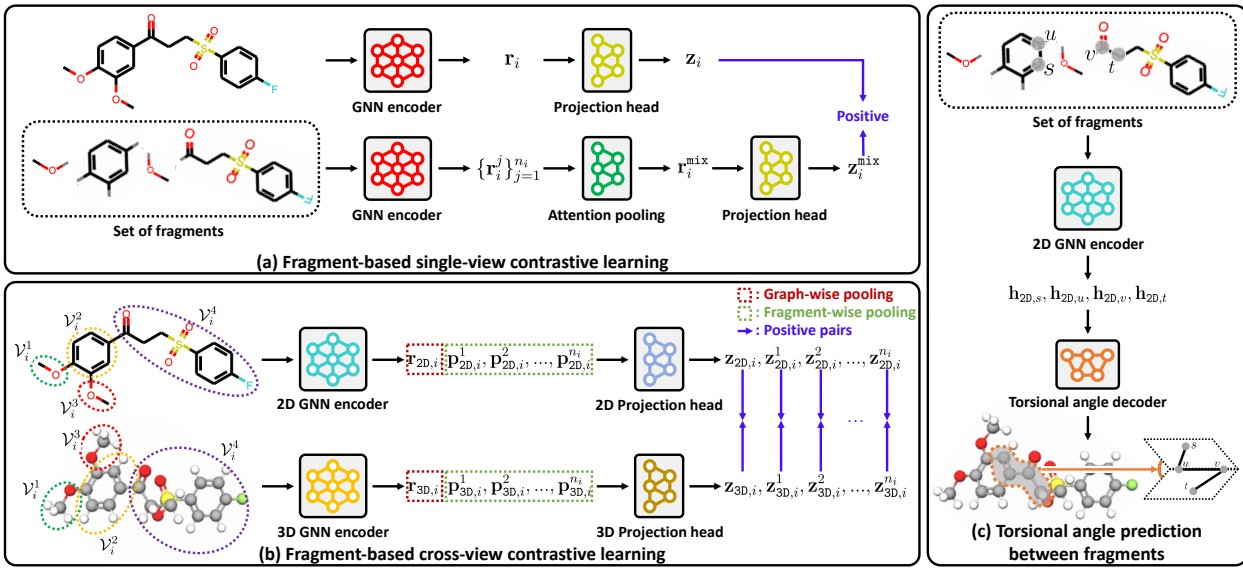

Figure 1: An overview of *Holistic Molecular representation learning via multi-view fragmentation (Holi-Mol)*. (a) Fragment-based single-view contrastive learning: a set of fragments is regarded as a positive view of the molecule. $\mathbf{r}_i$ and $\{\mathbf{r}_i^j\}_{j=1}^{n_i}$ refer to representations of the molecular graph $M_i$ and its fragments, respectively. (b) Fragment-based cross-view contrastive learning: positive pairs are constructed using 2D and 3D views of a molecule and its fragments. (c) Torsional angle prediction between fragments: 3D contextual information is learned by predicting the torsional angle.

Recently, graph neural networks (GNNs, Kipf & Welling, 2017) have gained much attention as a framework to learn the molecular graph representation due to its remarkable performance in predicting chemical properties (Wu et al., 2018). However, they often suffer from overfitting when the number of labeled training samples is insufficient (Rong et al., 2020b). To resolve this, researchers have investigated self-supervised learning that generates supervisory signals without labels to utilize a huge amount of unlabeled molecules (Rong et al., 2020a; Zhou et al., 2022).

The self-supervised learning approaches for 2D molecular graphs fall in two categories: (a) contrastive learning with graph augmentations, e.g., edge drop (Wang et al., 2021), and (b) masked substructure, e.g., motifs prediction (Hu et al., 2020a; Zhang et al., 2021). Specifically, the former (a) learns molecular representations by maximizing the agreement of similar positive views while minimizing that of dissimilar negative views on the representation space. Although this approach can learn global graph structures, it often fails to capture chemically meaningful local information (e.g., functional groups) since the graph augmentations may not preserve such substructures. In contrast, the latter approach (b) captures the local information well by predicting substructures directly from masked molecular graphs. Yet, it is unclear whether they learn the global information well.

In the field of chemistry, both global and local information in molecular graphs are chemically important. For example, long chain-like global molecular structures are associated with high water-octanol partition coefficients (Bhal, 2007), while the molecules with the fluorobenzene substructure are highly likely to be active on blood-brain barrier permeability (i.e., 97% active ratio in the BBBP dataset (Wu et al., 2018)). This inspires us to explore the following research question: *how to capture both global and local information for learning better molecular representations?*

**Contribution.** Our key idea is to (i) associate a global structure with a set of substructures and (ii) learn their relationships via contrastive learning to capture both global and local information. Specifically, we utilize *fragmentation* that decomposes a molecule into a set of chemically meaningful fragments, e.g., functional groups. In particular, we use the popular fragmentation strategy, Breaking of Retrosynthetically

Interesting Chemical Substructures (BRICS, Degen et al., 2008), which preserves the functional groups (Liu et al., 2017). We also utilize the 3D geometry[1] as another global and substructure views of contrastive learning and propose the 3D torsional angle prediction task between fragments. They enrich the molecular representations since such 3D information is related to various molecular properties, e.g., polarizability (Anslyn & Dougherty, 2006).

In an effort to jointly consider both global and local information acquired through fragmentation of the 2D and 3D multi-view unlabeled molecules, we introduce a novel molecular representation learning framework, **Holi**stic **Mol**ecular reprsentation learning via multi-view fragmentation (**Holi-Mol**). To be specific, Holi-Mol consists of the following components (see Figure 1).

(a) **Fragment-based single-view contrastive learning.** To facilitate interaction between a molecule and its substructures, we consider a set of fragments as a positive view of the molecule. We then apply contrastive learning between the whole molecular structure and the set of fragments for learning molecular representations via 2D and 3D molecule GNNs separately (i.e., single-view).

(b) **Fragment-based cross-view contrastive learning.** We consider molecule-level and fragment-level positive pairs between 2D topological and 3D geometric graphs. We then apply contrastive learning between the 2D and 3D views for learning 2D and 3D molecule GNNs jointly (i.e., cross-view). This cross-view objective enriches both global-level and local-level representations.

(c) **Torsional angle prediction between fragments.** To further exploit 3D geometric fragment-wise information, we suggest solving the torsional angle prediction task between adjacent fragments. Note that the torsional angle, defined by the dihedral angle of four adjacent atoms, is related to several 3D contextual properties, e.g., the energy surface around the bond (Smith, 2008).

For evaluation, we follow the multi-view pretraining setup (Stärk et al., 2022; Liu et al., 2022b): ($i$) pretrain a 2D molecule GNN using an unlabeled molecular dataset containing both 2D and 3D information, and then ($ii$) fine-tune the GNN on downstream tasks without 3D information. This setup using only 2D GNN is practically important since 3D information is often unattainable in downstream tasks due to the high computation cost. We explore the usefulness of our pre-trained 3D GNN in Appendix G.

Through extensive experiments, we demonstrate the superiority of our Holi-Mol framework over existing pretraining methods. Specifically, our GNN pretrained by Holi-Mol on the GEOM (Axelrod & Gomez-Bombarelli, 2022) dataset consistently outperforms the state-of-the-art method, GraphMVP (Liu et al., 2022b), when transferred to both MoleculeNet classification (Wu et al., 2018) and QM9 regression (Ramakrishnan et al., 2014) benchmarks (see Table 1 and 2, respectively). For example, we improve the average ROC-AUC score by $74.1 \rightarrow 75.5$ over the prior art on MoleculeNet. We further demonstrate the potential of Holi-Mol for other applications: semi-supervised/fully-supervised learning (see Table 3) and molecule retrieval (see Table 4).

## 2 Preliminaries

**Multi-view molecular representation learning.** The problem of interest in this paper is *molecular representation learning (MRL)* which aims to ($i$) learn a generalizable neural network for molecules $f : \mathcal{M} \rightarrow \mathbb{R}^d$ where $\mathcal{M}$ is the molecular space and $\mathbb{R}^d$ is the embedding space, and ($ii$) utilize the network $f$ for various chemical downstream tasks (e.g., toxicity prediction). Usually, ($i$) utilizes only unlabeled molecules (Sun et al., 2019; You et al., 2020; 2021; Liu et al., 2022b) due to the significant experimental cost to obtain label information of molecules compared to the cost of the collection of unlabeled molecules.

In general, a molecule $M \in \mathcal{M}$ can be represented by an attributed graph $M = (V, E, A, B, R)$ where $V$ is a set of nodes associated with atom features $A \in \mathbb{R}^{|V| \times d_{\text{atom}}}$ (e.g., atomic numbers), $E \subseteq V \times V$ is a set of edges associated with bond features $B \in \mathbb{R}^{|E| \times d_{\text{bond}}}$ (e.g., bond types), and $R \in \mathbb{R}^{|V| \times 3}$ is an array of 3D atom

---

[1]A molecule can be represented by (a) a 2D topological graph $(V, E)$ of nodes $V$ and edges $E$ or (b) a 3D geometric graph $(V, R)$ of nodes $V$ and 3D coordinates $R$.

positions. Conventionally, $M_{2D} = (V, E, A, B)$ and $M_{3D} = (V, A, R)$ are referred to 2D and 3D molecular graphs, respectively (Stärk et al., 2022; Liu et al., 2022b). We note that obtaining accurate 3D information $R$ is very expensive due to iterative quantum computations and thus many real-world applications suffer from the lack of such 3D information (Liu et al., 2022b).

To consider a wide range of downstream tasks, we focus on learning a graph neural network (GNN) for 2D molecular graphs $f_{2D} : \mathcal{M}_{2D} \to \mathbb{R}^d$ where $\mathcal{M}_{2D}$ is the 2D molecular graph space. To be specific, we (*i*) pretrain a 2D molecule GNN $f_{2D}$ using an unlabeled set of molecules $\mathcal{D}_u \subseteq \mathcal{M}$ containing both 2D and 3D information, and then (*ii*) fine-tune $f_{2D}$ on downstream tasks without 3D information, i.e., each task has a dataset $\mathcal{D} \subseteq \mathcal{M}_{2D} \times \mathcal{Y}$ where $\mathcal{Y}$ is the label space. Therefore, it is important to inject not only 2D topological information, but also 3D geometric information into the 2D molecule GNN $f_{2D}$ during pretraining. We remark that this *multi-view* pretraining setup has been recently investigated (Stärk et al., 2022; Liu et al., 2022b) to alleviate the scarcity of 3D topological information.

**2D molecule GNN** $f_{2D} : \mathcal{M}_{2D} \to \mathbb{R}^d$. For any 2D molecule $M_{2D} = (V, E, A, B) \in \mathcal{M}_{2D}$, GNNs for 2D molecules (2D-GNNs in short) compute molecular representations by applying (a) iterative neighborhood aggregation (i.e., message passing) to acquire node-level representations based on the graph $(V, E)$ and then (b) a readout function (e.g., mean pooling) to create graph-level representations. Formally, node- and graph-level representations of $L$-layer 2D-GNN are as follows:

$$\mathbf{h}_v^{(\ell)} := \mathtt{MP}(\mathbf{h}_v^{(\ell-1)}, \{\mathbf{h}_u^{(\ell-1)}, B_{uv}\}_{u \in \mathcal{N}(v)}), \ \ell \in [L],$$

$$f_{2D}(M) := f_{2D}(M_{2D}) = \mathtt{Readout}(\{\mathbf{h}_v^{(L)}\}_{v \in V}),$$

where $\mathtt{MP}(\cdot)$ is a message passing layer, $\mathtt{Readout}(\cdot)$ is a readout function, $\mathbf{h}_v^{(0)} = A_v$ is the atom feature for a node $v$, $B_{uv}$ is the bond feature for an edge $(u, v) \in E$, and $\mathcal{N}(v)$ is the set of adjacent nodes of $v$. In this work, we mainly use the graph isomorphism network (GIN) architecture (Xu et al., 2019) following the standard MRL setup (Hu et al., 2020a).

**3D molecule GNN** $f_{3D} : \mathcal{M}_{3D} \to \mathbb{R}^d$. For any 3D molecule $M_{3D} = (V, A, R) \in \mathcal{M}_{3D}$, GNNs for 3D molecules (3D-GNNs in short) compute molecular representations by applying (a) iterative geometric interactions through distances and angles between nodes (i.e., atoms) to acquire node-level representations based on the 3D geometry $R$ and then (b) a readout function to create graph-level representation Formally, node- and graph- level representations of $L$-layer 3D-GNN are as follows:

$$\mathbf{h}_v^{(\ell)} := \mathtt{IB}(\mathbf{h}_v^{(\ell-1)}, R_v, \{\mathbf{h}_u^{(\ell-1)}, R_u\}_{u \in V \setminus \{v\}}), \ell \in [L],$$

$$f_{3D}(M) := f_{3D}(M_{3D}) = \mathtt{Readout}(\{\mathbf{h}_v^{(L)}\}_{v \in V}),$$

where $\mathtt{IB}(\cdot)$ is an interaction block, $\mathtt{Readout}(\cdot)$ is a readout function, $\mathbf{h}_v^{(0)} = A_v$ and $R_v$ is the atom feature and the 3D position for a node $v$, respectively. In this work, we mainly use the SchNet architecture (Schütt et al., 2017) following the setup of Liu et al. (2022b).

**Contrastive learning.** Contrastive learning aims to learn discriminative representations by attracting positive views while repelling negative views on the representation space, e.g., see Chen et al. (2020). A common practice for generating positive views is to utilize semantic-preserving transformations. Let $(\mathbf{x}, \mathbf{x}^+)$ be a positive pair generated by the transformations and $(\mathbf{x}, \mathbf{x}^-)$ be a negative pair obtained from different instances in a mini-batch. If $\mathbf{z}, \mathbf{z}^+$, and $\mathbf{z}^-$ are the representations of $\mathbf{x}, \mathbf{x}^+$, and $\mathbf{x}^-$, respectively, then the contrastive learning objective $\mathcal{L}_{CL}$ can be written as follows (You et al., 2020; Chen et al., 2020):

$$\mathcal{L}_{CL}(\mathbf{z}, \mathbf{z}^+, \{\mathbf{z}^-\}) = -\log \frac{\exp(\mathrm{sim}(\mathbf{z}, \mathbf{z}^+)/\tau)}{\sum_{\mathbf{z}^-} \exp(\mathrm{sim}(\mathbf{z}, \mathbf{z}^-)/\tau)}, \tag{1}$$

where $\mathrm{sim}(\mathbf{z}, \widetilde{\mathbf{z}}) = \mathbf{z}^\top \widetilde{\mathbf{z}} / \|\mathbf{z}\|_2 \|\widetilde{\mathbf{z}}\|_2$ is the cosine similarity and $\tau$ is a temperature-scaler. Here, the set $\{\mathbf{z}^-\}$ may include the positive $\mathbf{z}^+$ depending on the choice of objectives, e.g., NT-Xent (Chen et al., 2020).

## 3 Holistic molecular representation learning via multi-view fragmentation

In this section, we propose a holistic molecular representation learning via multi-view fragmentation, coined Holi-Mol, which fully exploits both global and local molecular information using 2D and 3D multi-view

fragmentation in order to learn good representation of 2D-GNN $f_{\text{2D}}$. First, we introduce our fragmentation scheme (Section 3.1). Thereafter, we describe details of our fragment-based components: single-view contrastive learning (Section 3.2), cross-view contrastive learning (Section 3.3), and torsional angle prediction between fragments (Section 3.4). Our framework is illustrated in Figure 1.

### 3.1 Molecular fragmentation

Our framework relies on the molecular fragmentation which decomposes a molecule into a set of chemically meaningful fragments (i.e., substructures). In this paper, we mainly use the popular strategy, Breaking of Retrosynthetically Interesting Chemical Substructures (BRICS, Degen et al., 2008), which is designed to preserve most chemically informative substructures and has been widely adopted in the prior literatures (Zhang et al., 2021; Yang et al., 2022). The efficacy of BRICS is further verified through an analysis in Appendix H. Formally, for a molecule $M = (V, E, A, B, R)$, a fragmentation scheme decomposes $M$ into a set of fragments $\{M^j\}$ where $M^j = (V^j, E^j, A^j, B^j, R^j)$ is the $j$-th connected component of the graph induced by the fragmentation. We refer 2D and 3D fragment as $M_{\text{2D}}^j$ and $M_{\text{3D}}^j$, respectively.

### 3.2 Fragment-based contrastive learning: Single-view objective

To incorporate both global and local information into molecular representation, we propose contrastive learning objective upon molecular fragmentation. To achieve this, we first utilize fragmentation (see Section 3.1). Then, Holi-Mol learns the relationship between molecule-level and fragment-level information by considering a set of fragments as a positive view of a molecule. To be specific, given a training batch $\{M_i\}_{i=1}^n$, we consider $(M_i, \{M_i^j\}_{j=1}^{n_i})$ as a positive pair (i.e., they share the same chemical semantics) where $n_i$ is the number of fragments of the molecule $M_i$. To aggregate representations of $\{M_i^j\}_{j=1}^{n_i}$, we use the attention pooling (Li et al., 2016). Formally, the representation for the set $\{M_i^j\}_{j=1}^{n_i}$ is:

$$\mathbf{r}_i^{\texttt{mix}} := \sum_{j=1}^{n_i} \frac{\exp(\mathbf{a}^\top \mathbf{r}_i^j + b)}{\sum_{l=1}^{n_i} \exp(\mathbf{a}^\top \mathbf{r}_i^l + b)} \cdot \mathbf{r}_i^j,$$

where $\mathbf{r}_i^j := f(M_i^j)$ is the representation for each fragment, $\mathbf{a} \in \mathbb{R}^d$ and $b \in \mathbb{R}$ are learnable parameters. Similarly, we compute the molecular representation $\mathbf{r}_i = f(M_i)$ for the whole structure. Then, we separately optimize the 2D-GNN $f_{\text{2D}}$ and the 3D-GNN $f_{\text{3D}}$ along with projection heads $g_{\text{2D (or 3D)}} : \mathbb{R}^d \to \mathbb{R}^d$ by the following contrastive objective with the fragment-based positive pairs:

$$\mathcal{L}_{\texttt{single}} := \frac{1}{n} \sum_{i=1}^n \left( \mathcal{L}_{\text{CL}}(\mathbf{z}_{\text{2D},i}, \mathbf{z}_{\text{2D},i}^{\texttt{mix}}, \{\mathbf{z}_{\text{2D},j}^{\texttt{mix}}\}_{j \neq i}) + \mathcal{L}_{\text{CL}}(\mathbf{z}_{\text{3D},i}, \mathbf{z}_{\text{3D},i}^{\texttt{mix}}, \{\mathbf{z}_{\text{3D},j}^{\texttt{mix}}\}_{j \neq i}) \right), \tag{2}$$

where $\mathbf{z}_i = g(\mathbf{r}_i)$ and $\mathbf{z}_i^{\texttt{mix}} = g(\mathbf{r}_i^{\texttt{mix}})$.

### 3.3 Fragment-based contrastive learning: Cross-view objective

To bring 3D geometric information into molecular representation of 2D-GNN effectively, we propose cross-view contrastive objective. Since these views provide different chemical information (e.g., atom-bond connectivity (Estrada et al., 1998) and energy surface (Smith, 2008) in the 2D and 3D views, respectively), such a cross-view objective could make the representation $f(M)$ more informative. To further utilize global and local information of 3D molecules, we use both of molecule-level and fragment-level cross-view contrastive learning objectives into our framework.

**Molecule-level objective.** Here, we consider $(M_{\text{2D},i}, M_{\text{3D},i})$ as a positive pair. Then, the molecule-level contrastive objective can be written as follows:

$$\mathcal{L}_{\texttt{cross,mol}} := \frac{1}{2n} \sum_{i=1}^n \left( \mathcal{L}_{\text{CL}}(\mathbf{z}_{\text{2D},i}, \mathbf{z}_{\text{3D},i}, \{\mathbf{z}_{\text{3D},k}\}_{k=1}^n) + \mathcal{L}_{\text{CL}}(\mathbf{z}_{\text{3D},i}, \mathbf{z}_{\text{2D},i}, \{\mathbf{z}_{\text{2D},k}\}_{k=1}^n) \right). \tag{3}$$

This objective is well-known to be effective in general multi-view representation learning (Radford et al., 2021; Tian et al., 2020), and also investigated in molecular representation learning (Stärk et al., 2022; Liu et al., 2022b). However, modeling the cross-view contrastive objective based solely on the similarity of molecule-level representations may lack capturing fragment-level information (i.e., chemical property at a finer level). Therefore, inspired by token-wise contrastive learning in the vision-language domain (Yao et al., 2022), we suggest *fragment-level cross-view contrastive learning* in what follows.

**Fragment-level objective.** We consider $(M_{\text{2D},i}^j, M_{\text{3D},i}^j)$ as a fragment-level positive pair and consider $(M_{\text{2D},i}^j, M_{\text{3D},k}^m)$ as fragment-level negative pairs for $k \neq i$. To achieve this, we consider *context-aware* fragment representations $\mathbf{p}_i^j$, to incorporate the neighboring context of the fragments.[2] To be specific, we define the $j$-th fragment representation $\mathbf{p}_i^j$ of a molecule $M_i$ via fragment-wise pooling as follows:

$$\mathbf{p}_i^j := \frac{1}{|V_i^j|} \sum_{v \in V_i^j} \mathbf{h}_{v,i}, \tag{4}$$

where $\{\mathbf{h}_{v,i}\}_{v \in V}$ are the last-layer node representations of the whole molecular structure $M_i$ obtained by a GNN $f$, and $V_i^j$ is a set of nodes in the $j$-th fragment. Here, $p_i^j$ contains a global context of a molecule since $h_{v,i}$ is obtained by message passing or interaction block. We then compute latent fragment representations by a projector $g$, e.g., $\mathbf{z}_{\text{2D},i}^j = g_{\text{2D}}(\mathbf{p}_{\text{2D},i}^j)$. Using these representations, we define the average of fragment-wise similarities $s_{i,i}$ in molecule $M_i$ and $s_{i,k}$ between molecules $M_i$ and $M_k$ as follows:

$$s_{i,i} := \frac{1}{n_i} \sum_{l=1}^{n_i} \text{sim}(\mathbf{z}_{\text{2D},i}^l, \mathbf{z}_{\text{3D},i}^l), \text{ and}$$

$$s_{i,k}^{\text{2D (or 3D)}} := \frac{1}{n_i} \sum_{l=1}^{n_i} \max_{1 \leq m \leq n_k} \text{sim}(\mathbf{z}_{\text{2D (or 3D)},i}^l, \mathbf{z}_{\text{3D (or 2D)},k}^m),$$

where $n_i, n_k$ is the number of fragments of the molecule $M_i, M_k$, respectively. Finally, we formulate our fragment-level cross-view contrastive objective as follows:

$$\mathcal{L}_{\text{cross,frag}} := -\frac{1}{2n} \sum_{i=1}^n \left( \log \frac{e^{s_{i,i}/\tau}}{e^{s_{i,i}/\tau} + \sum_{k \neq i} e^{s_{i,k}^{\text{2D}}/\tau}} + \log \frac{e^{s_{i,i}/\tau}}{e^{s_{i,i}/\tau} + \sum_{k \neq i} e^{s_{i,k}^{\text{3D}}/\tau}} \right). \tag{5}$$

To sum up, our cross-view objective is as follows:

$$\mathcal{L}_{\text{cross}} := \frac{1}{2} \left( \mathcal{L}_{\text{cross,mol}} + \mathcal{L}_{\text{cross,frag}} \right).$$

### 3.4 Torsional angle prediction between fragments

To further incorporate the 3D geometry into 2D-GNN $f_{\text{2D}}$, we propose an additional pretext task, where $f_{\text{2D}}$ learns to predict torsional angles around the fragmented bonds given fragments of 2D molecular graphs $\{M_{\text{2D},i}^j\}_{j=1}^{n_i}$. A torsional angle is defined by an atom quartet $(s, u, v, t)$ which is a sequence of four atoms, each pair connected by a covalent bond. As illustrated in Figure 1(c), the torsional angle is the angle between the planes defined by $(s, u, v)$ and $(u, v, t)$, and it encodes important 3D local properties, e.g., energy surface around the atoms (Smith, 2008). Thus, by predicting the torsional angle from any two arbitrary fragments of a molecule $M$, where they are originally connected by the bond $(u, v)$, the 2D-GNN $f_{\text{2D}}$ is expected to learn the 3D contextual properties around the fragmented bond.

We now define the torsional angle prediction task for each fragmented bond: for a 2D molecule $M_{\text{2D},i}$ and a fragmented bond $(u, v) \in E_{\text{2D},i}$, we randomly select non-hydrogen atoms $s$ and $t$ adjacent to $u$ and $v$,

---

[2]Contrary to the fragment representations proposed in Section 3.2, which does not incorporate neighboring context nearby the fragments, we now consider context-aware fragment representations. These representations are obtained by fragment-wise pooling after message passing (see Eq. (4)).

Table 1: Test ROC-AUC score on the MoleculeNet downstream molecular property classification benchmarks. We employ GIN (Xu et al., 2019) as the 2D-GNN architecture and pretrain with 50k molecules from the GEOM dataset (Axelrod & Gomez-Bombarelli, 2022), following Liu et al. (2022b). We report mean and standard deviation over 3 different seeds. We mark the best mean score and scores within one standard deviation of the best mean score to be bold. We denote the scores obtained from Liu et al. (2022b) with (*). Otherwise, we reproduce scores under the same setup. Scores obtained through fine-tuning of the officially provided checkpoints are denoted by (†).[3]

| Methods | BBBP | Tox21 | ToxCast | Sider | Clintox | MUV | HIV | Bace | Avg. |
|---|---|---|---|---|---|---|---|---|---|
| - | $65.4_{\pm2.4}$ | $\mathbf{74.9}_{\pm0.8}$ | $61.6_{\pm1.2}$ | $58.0_{\pm2.4}$ | $58.8_{\pm5.5}$ | $71.0_{\pm2.5}$ | $\mathbf{75.3}_{\pm0.5}$ | $72.6_{\pm4.9}$ | 67.2 |
| Pretrained with 50k 2D molecular graphs of GEOM and fine-tuned on 2D molecular graphs of MoleculeNet | | | | | | | | | |
| AttrMask* (Hu et al., 2020a) | $70.2_{\pm0.5}$ | $74.2_{\pm0.8}$ | $62.5_{\pm0.4}$ | $60.4_{\pm0.6}$ | $68.6_{\pm9.6}$ | $73.9_{\pm1.3}$ | $74.3_{\pm1.3}$ | $77.2_{\pm1.4}$ | 70.2 |
| GPT-GNN* (Hu et al., 2020b) | $64.5_{\pm1.1}$ | $\mathbf{75.3}_{\pm0.5}$ | $62.2_{\pm0.1}$ | $57.5_{\pm4.2}$ | $57.8_{\pm3.1}$ | $\mathbf{76.1}_{\pm2.3}$ | $75.1_{\pm0.2}$ | $77.6_{\pm0.5}$ | 68.3 |
| Infomax* (Sun et al., 2019) | $69.2_{\pm0.8}$ | $73.0_{\pm0.7}$ | $62.0_{\pm0.3}$ | $59.2_{\pm0.2}$ | $75.1_{\pm5.0}$ | $74.0_{\pm1.5}$ | $74.5_{\pm1.8}$ | $73.9_{\pm2.5}$ | 70.1 |
| ContextPred* (Hu et al., 2020a) | $\mathbf{71.2}_{\pm0.9}$ | $73.3_{\pm0.5}$ | $62.8_{\pm0.3}$ | $59.3_{\pm1.4}$ | $73.7_{\pm4.0}$ | $72.5_{\pm2.2}$ | $\mathbf{75.8}_{\pm1.1}$ | $78.6_{\pm1.4}$ | 70.9 |
| GraphLoG* (Xu et al., 2021) | $67.8_{\pm1.7}$ | $73.0_{\pm0.3}$ | $62.2_{\pm0.4}$ | $57.4_{\pm2.3}$ | $62.0_{\pm1.8}$ | $73.1_{\pm1.7}$ | $73.4_{\pm0.6}$ | $78.8_{\pm0.7}$ | 68.5 |
| G-Contextual* (Rong et al., 2020a) | $70.3_{\pm1.6}$ | $\mathbf{75.2}_{\pm0.3}$ | $62.6_{\pm0.3}$ | $58.4_{\pm0.6}$ | $59.9_{\pm8.2}$ | $72.3_{\pm0.9}$ | $\mathbf{75.9}_{\pm0.9}$ | $79.2_{\pm0.3}$ | 69.2 |
| G-Motif* (Rong et al., 2020a) | $66.4_{\pm3.4}$ | $73.2_{\pm0.8}$ | $62.6_{\pm0.5}$ | $60.6_{\pm1.1}$ | $77.8_{\pm2.0}$ | $73.3_{\pm2.0}$ | $73.8_{\pm1.4}$ | $73.4_{\pm4.0}$ | 70.1 |
| GraphCL* (You et al., 2020) | $67.5_{\pm3.3}$ | $\mathbf{75.0}_{\pm0.3}$ | $62.8_{\pm0.2}$ | $60.1_{\pm1.3}$ | $78.9_{\pm4.2}$ | $\mathbf{77.1}_{\pm1.0}$ | $75.0_{\pm0.4}$ | $68.7_{\pm7.8}$ | 70.1 |
| JOAO* (You et al., 2021) | $66.0_{\pm0.6}$ | $74.4_{\pm0.7}$ | $62.7_{\pm0.6}$ | $60.7_{\pm1.0}$ | $66.3_{\pm3.9}$ | $\mathbf{77.0}_{\pm2.2}$ | $76.6_{\pm0.5}$ | $72.9_{\pm2.0}$ | 70.6 |
| MGSSL (Zhang et al., 2021) | $67.3_{\pm0.9}$ | $74.5_{\pm0.2}$ | $63.6_{\pm0.4}$ | $58.4_{\pm0.2}$ | $75.4_{\pm3.8}$ | $73.9_{\pm1.4}$ | $77.2_{\pm2.5}$ | $76.2_{\pm1.3}$ | 70.8 |
| MolCLR (Wang et al., 2021) | $67.6_{\pm0.6}$ | $74.4_{\pm1.3}$ | $62.9_{\pm0.2}$ | $58.7_{\pm1.1}$ | $57.9_{\pm3.0}$ | $70.8_{\pm2.8}$ | $\mathbf{75.4}_{\pm1.2}$ | $74.6_{\pm3.5}$ | 67.8 |
| D-SLA (Kim et al., 2022) | $69.6_{\pm2.4}$ | $73.7_{\pm0.7}$ | $63.3_{\pm0.2}$ | $59.2_{\pm2.0}$ | $60.5_{\pm1.0}$ | $75.3_{\pm0.6}$ | $\mathbf{75.8}_{\pm0.9}$ | $\mathbf{81.2}_{\pm2.5}$ | 69.8 |
| Mole-BERT (Xia et al., 2023) | $69.8_{\pm1.7}$ | $\mathbf{74.9}_{\pm0.5}$ | $63.7_{\pm0.6}$ | $58.5_{\pm0.6}$ | $80.5_{\pm2.4}$ | $72.5_{\pm2.4}$ | $\mathbf{75.9}_{\pm1.2}$ | $78.4_{\pm1.6}$ | 71.8 |
| Pretrained with 50k 2D and 3D molecular graphs of GEOM and fine-tuned on 2D molecular graphs of MoleculeNet | | | | | | | | | |
| 3D-InfoMax (Stärk et al., 2022) | $67.9_{\pm1.2}$ | $\mathbf{75.3}_{\pm0.3}$ | $\mathbf{64.6}_{\pm0.4}$ | $59.6_{\pm0.7}$ | $89.7_{\pm0.5}$ | $\mathbf{76.7}_{\pm0.6}$ | $73.4_{\pm1.2}$ | $79.9_{\pm0.9}$ | 73.4 |
| GraphMVP† (Liu et al., 2022b) | $69.6_{\pm0.2}$ | $\mathbf{75.6}_{\pm0.7}$ | $63.7_{\pm0.3}$ | $61.3_{\pm0.6}$ | $89.0_{\pm1.4}$ | $75.7_{\pm1.0}$ | $75.1_{\pm0.3}$ | $\mathbf{80.9}_{\pm1.3}$ | 73.9 |
| GraphMVP-G† (Liu et al., 2022b) | $70.1_{\pm0.7}$ | $\mathbf{75.3}_{\pm0.9}$ | $64.2_{\pm0.9}$ | $61.0_{\pm0.5}$ | $89.4_{\pm1.5}$ | $\mathbf{77.7}_{\pm1.6}$ | $75.3_{\pm0.8}$ | $80.2_{\pm1.5}$ | 74.1 |
| GraphMVP-C† (Liu et al., 2022b) | $69.6_{\pm1.4}$ | $74.6_{\pm0.1}$ | $64.1_{\pm0.2}$ | $\mathbf{63.0}_{\pm0.1}$ | $88.7_{\pm2.6}$ | $73.9_{\pm1.7}$ | $74.7_{\pm2.0}$ | $\mathbf{81.3}_{\pm0.7}$ | 73.7 |
| **Holi-Mol (Ours)** | $\mathbf{71.4}_{\pm0.4}$ | $75.2_{\pm0.7}$ | $\mathbf{65.1}_{\pm0.8}$ | $61.0_{\pm0.6}$ | $\mathbf{95.2}_{\pm1.0}$ | $\mathbf{77.6}_{\pm1.0}$ | $\mathbf{76.3}_{\pm0.4}$ | $\mathbf{82.3}_{\pm1.6}$ | 75.5 |

respectively, and compute the torsional angle $y$ of the quartet $(s, u, v, t)$ on the molecule $M_i$. If $\mathcal{T}$ is a collection of the tasks for all fragments, our loss function can be written as follows:

$$\mathcal{L}_{\mathtt{tor}} := \frac{1}{|\mathcal{T}|} \sum_{(i,s,u,v,t,y)\in\mathcal{T}} \mathcal{L}_{\mathtt{CE}}(\hat{y}_i(s,u,v,t), y), \tag{6}$$

where $\mathcal{L}_{\mathtt{CE}}$ is the cross-entropy loss, $y$ is the binned label for the angle, and $\hat{y}_i(s,u,v,t) := g_{\mathtt{tor}}([\mathbf{h}_{\mathtt{2D},a,i}]_{a\in\{s,u,v,t\}})$ is the prediction from the concatenation of node representations of atoms $(s, u, v, t)$ of the molecule $M_{\mathtt{2D},i}$ using a multi-layer perceptron (MLP) $g_{\mathtt{tor}}(\cdot)$.

### 3.5 Overall training objective

By aggregating the objectives proposed in Section 3.2, 3.3, and 3.4, we propose our total training loss function. In summary, we consider a set of fragments as a positive view of a molecule while maximizing the consistency of the outputs of 2D and 3D-GNNs both at the molecule-level and at the fragment-level. Additionally, we incorporate 3D contextual information by predicting the torsional angles around the fragmented bonds. The overall loss function is as follows:

$$\mathcal{L}_{\mathtt{Holi-Mol}} := \mathcal{L}_{\mathtt{single}} + \mathcal{L}_{\mathtt{cross}} + \mathcal{L}_{\mathtt{tor}}. \tag{7}$$

Note that $\tau$ in Eq. (1) and (5) is the only hyperparameter we introduced. We set $\tau = 0.1$ following You et al. (2020).

---

[3]GraphMVP (Liu et al., 2022b) pretrains with explicit hydrogens, but fine-tunes without explicit hydrogens. We report fine-tuning results with explicit hydrogens from official checkpoints. Thus, our reported average value is slightly higher than the original paper.

Table 2: Test MAE score on the QM9 downstream quantum property regression benchmarks. For ours and all baselines, we employ GIN (Xu et al., 2019) as the 2D-GNN architecture and pretrain with entire 310k molecules from the GEOM dataset (Axelrod & Gomez-Bombarelli, 2022). We mark the best score bold.

| Methods | ZPVE ↓ | $\mu$ ↓ | $\alpha$ ↓ | $C_v$ ↓ | LUMO ↓ | HOMO ↓ | $\varepsilon_{gap}$ ↓ | $R^2$ ↓ | $U_0$ ↓ | $U_{298}$ ↓ | $H_{298}$ ↓ | $G_{298}$ ↓ |
|---|---|---|---|---|---|---|---|---|---|---|---|---|
| - | 43.7 | 0.059 | 0.400 | 0.144 | 80.5 | 89.4 | 171.0 | 3.27 | 62.9 | 61.8 | 57.0 | 48.1 |
| *Pretrained on 310k 2D molecular graphs of GEOM and fine-tuned on 2D molecular graphs of QM9* | | | | | | | | | | | | |
| 3D-Infomax Stärk et al. (2022) | 27.0 | 0.051 | 0.355 | 0.126 | 63.4 | 55.2 | 103.8 | 2.99 | **38.8** | 45.6 | 41.0 | 40.8 |
| GraphMVP-G Liu et al. (2022b) | 24.1 | 0.051 | 0.367 | 0.123 | 59.1 | 53.8 | 100.4 | 2.97 | 39.9 | 44.2 | 41.0 | 40.3 |
| **Holi-Mol (Ours)** | **24.0** | **0.049** | **0.353** | **0.121** | **57.1** | **51.8** | **97.1** | **2.90** | 39.2 | **42.9** | **40.3** | **40.0** |

Table 3: Test MAE score of semi-supervised learning on the QM9 downstream quantum property regression benchmarks. We employ GIN (Xu et al., 2019) as the 2D-GNN architecture and pretrain with the 110k QM9 training dataset. Then we fine-tune across different label fraction of the QM9 training dataset. We mark the best score bold.

| Methods | ZPVE ↓ | | | LUMO ↓ | | | HOMO ↓ | | | $U_0$ ↓ | | |
|---|---|---|---|---|---|---|---|---|---|---|---|---|
| Label Fraction (%) | 20 | 50 | 100 | 20 | 50 | 100 | 20 | 50 | 100 | 20 | 50 | 100 |
| - | 111.0 | 87.1 | 43.7 | 236.0 | 140.6 | 80.5 | 233.6 | 128.1 | 89.4 | 165.5 | 82.8 | 62.9 |
| *Pretrained on 110k 2D and 3D molecular graphs of QM9 and fine-tuned on 2D molecular graphs of QM9* | | | | | | | | | | | | |
| 3D-Infomax Stärk et al. (2022) | 87.2 | 42.8 | 24.4 | 215.0 | 98.4 | 57.9 | 181.0 | 102.4 | 57.7 | 148.2 | 75.0 | 42.1 |
| GraphMVP-G Liu et al. (2022b) | 85.4 | 42.8 | 24.4 | 214.3 | 99.7 | 59.7 | 177.3 | 100.0 | 56.9 | 145.7 | 74.5 | 42.2 |
| **Holi-Mol (Ours)** | **83.7** | **39.4** | **22.2** | **202.2** | **97.8** | **54.6** | **172.9** | **91.0** | **48.4** | **138.7** | **71.8** | **38.0** |

# 4 Experiments

In this section, we extensively compare Holi-Mol with the existing molecular graph representation learning methods. We evaluate Holi-Mol and baselines on various downstream molecular property prediction tasks after pretraining on (unlabeled) molecular dataset. For an extensive evaluation in various downstream setups, we consider (1) transfer-learning, i.e., pre-training and fine-tuning distribution are different (Table 1 and 2), and (2) semi-supervised learning, i.e., pre-training and fine-tuning distribution are the same (Table 3). We further discuss the evaluation setups in Appendix O. Also, we perform an ablation study to investigate the effect of each component of Holi-Mol for discriminating molecules.

## 4.1 Experimental setup

**Baselines.** We consider recently proposed multi-view molecular representation learning methods which utilize both 2D and 3D molecular graphs in pretraining: 3D-Infomax (Stärk et al., 2022), GraphMVP, and GraphMVP-{G,C} (Liu et al., 2022b). Following GraphMVP, we also compare with molecular representation learning methods which pretrain solely with 2D molecular graphs, including the recently proposed MGSSL (Zhang et al., 2021), MolCLR (Wang et al., 2021), and D-SLA (Kim et al., 2022). We provide more details on baselines in Appendix B.

**Datasets.** For pretraining, we consider the GEOM (Axelrod & Gomez-Bombarelli, 2022) and the QM9 (Ramakrishnan et al., 2014) datasets, which consist of 2D and 3D paired molecular graphs. We consider (a) transfer learning on the binary classification tasks from the MoleculeNet benchmark (Wu et al., 2018), and (b) transfer learning and semi-supervised learning on the regression tasks using QM9 (Ramakrishnan et al., 2014). Following Liu et al. (2022b), we use *scaffold split* (Chen et al., 2012) in MoleculeNet experiments which splits the molecules based on their substructures. For QM9 experiments, we follow the setup of Liu et al. (2021) which splits the dataset into 110,000 molecules for training, 10,000 molecules for validation, and 10,831 molecules for test. Detailed explanation about datasets can be found in Appendix D.

Table 4: Visualization of the molecules retrieved by MGSSL (Zhang et al., 2021), GraphMVP-G (Liu et al., 2022b), and Holi-Mol (Ours). We report top-2 closest molecules from the Tox21 dataset with respect to the query molecule in terms of cosine similarity in representation space. We utilize models pretrained with the GEOM dataset (Axelrod & Gomez-Bombarelli, 2022). We mark BRICS fragments as dotted lines.

**Architectures.** We employ 5-layer graph isomorphism network (GIN) (Xu et al., 2019) as 2D-GNN $f_{\text{2D}}$ and 6-layer SchNet (Schütt et al., 2017) as 3D-GNN $f_{\text{3D}}$. The configuration is drawn from GraphMVP (Liu et al., 2022b) for a fair comparison.

**Hardwares.** We use a single NVIDIA GeForce RTX 3090 GPU with 36 CPU cores (Intel(R) Core(TM) i9-10980XE CPU @ 3.00GHz) for self-supervised pretraining, and a single NVIDIA GeForce RTX 2080 Ti GPU with 40 CPU cores (Intel(R) Xeon(R) CPU E5-2630 v4 @ 2.20GHz) for fine-tuning.

## 4.2 Main results

**MoleculeNet classification task.** As reported in Table 1, Holi-Mol achieves the best average test ROC-AUC score when transferred to the MoleculeNet (Wu et al., 2018) downstream tasks after pretrained with 50k molecules from the GEOM (Axelrod & Gomez-Bombarelli, 2022) dataset. For example, Holi-Mol outperforms the recently proposed method for 2D molecular graph, Mole-BERT (Xia et al., 2023), in all downstream tasks. Furthermore, Holi-Mol improves the best average ROC-AUC score baseline, GraphMVP-G (Liu et al., 2022b), by 74.1 → 75.5, achieving the state-of-the-art performance on 7 out of 8 downstream tasks. We emphasize that the improvement of Holi-Mol is consistent over the downstream tasks. For example, GraphMVP-C (Liu et al., 2022b) achieves the best performance on Sider, while it fails to generalize on Tox21, resulting in even lower ROC-AUC score compared to the model without pretraining. On the other hand, Holi-Mol shows the best average performance with no such a failure case, i.e., Holi-Mol learns well-generalizable representations over several downstream tasks. We provide further analysis in Appendix E and F.

**QM9 regression task.** Table 2 and 3 show the overall results of transfer learning and semi-supervised learning on the QM9 (Ramakrishnan et al., 2014) regression benchmarks, respectively. For transfer learning (Table 2), we pretrain with 310k molecules from the GEOM (Axelrod & Gomez-Bombarelli, 2022) dataset. Holi-Mol outperforms the baselines, achieving the best performances on 11 out of 12 downstream tasks. We emphasize that Holi-Mol outperforms the baselines when transferred to both MoleculeNet and QM9 downstream tasks. For semi-supervised learning (Table 3), Holi-Mol successfully improves the baselines over all tasks and label fractions. In particular, Holi-Mol shows superior performance even in the fully supervised learning scenario (i.e., 100% label fraction), e.g., 89.4 → 48.4 for HOMO. This implies that Holi-Mol indeed finds "good initialization" of GNN and shows its wide applicability. More experimental results on the QM9 dataset can be found in Appendix G and N. We provide the regression results on PCQM4Mv2 in Appendix K.

**Molecule retrieval.** In Table 4, we further perform molecule retrieval task for qualitative analysis. Using pretrained models by MGSSL (Zhang et al., 2021) and GraphMVP-G (Liu et al., 2022b), and Holi-Mol, we visualize the molecules in the Tox 21 dataset, which have similar representations with the query molecule based on the cosine similarity. We observe that the state-of-the-art method, GraphMVP-G, does not find molecules with similar fragments to the query molecules. While MGSSL leverages local information of molecules in its training scheme and partially finds local substructure (indicated by dotted lines in Table 4) of molecules, Holi-Mol effectively retrieves molecules with common fragments and similar global structures of the query molecules. We further provide a t-SNE plot regarding the obtained representations in Appendix J.

Table 5: Average ROC-AUC score with different positive view construction strategy across 8 downstream tasks in MoleculeNet.

| Positive view construction | Fragmentation strategy | Avg. |
|---|---|---|
| Nodedrop, Subgraph (You et al., 2020) | - | 73.4 |
| **A set of fragments (Ours)** | Random bond deletion | 73.5 |
| | Random non-ring bond deletion | 74.0 |
| | **BRICS decomposition (Ours)** | **75.5** |

Table 6: Effectiveness of each objective as measured on the average ROC-AUC score across 8 downstream tasks in MoleculeNet.

| Pretraining data | Cross-view interaction level | | | Avg. |
|---|---|---|---|---|
| | Molecule | Fragment | Torsion | |
| Single-view (2D) | - | - | - | 72.4 |
| Multi-view (2D&3D) | ✓ | - | - | 74.7 |
| | ✓ | ✓ | - | 75.1 |
| | ✓ | ✓ | ✓ | **75.5** |

### 4.3 Ablation study

**Fragment-based positive view construction.** In Table 5, we investigate how our positive view construction strategy is effective. We first compare our strategy with the alternative: using an augmented molecular graph (i.e., random subgraph) as a positive view (You et al., 2020). We observe that deleting random bonds for positive-view construction does not improve the performance ($73.4 \to 73.5$), since important substructures of molecules (e.g., aromatic ring) can be easily broken by random deletion of bonds, which could lead to significant change in chemical properties. Preventing such ring deformation increases overall performance by $73.5 \to 74.0$. BRICS decomposition further incorporates chemical prior to obtain semantic-preserved local information, boosting the performance by $74.0 \to 75.5$. The result implies that considering chemically informative substructures is a key component of our framework. We provide detailed results in Appendix H.

**Effectiveness of multi-view pretraining.** In Table 6, we evaluate the impact of individual objectives on our total loss $\mathcal{L}_{\texttt{Holi-Mol}}$. We observe that molecule-level cross-view contrastive learning ($\mathcal{L}_{\texttt{cross,mol}}$; Eq. (3)) between 2D and 3D molecular views improves the overall performance by $72.4 \to 74.7$. Introducing fragment-level cross-view contrastive learning ($\mathcal{L}_{\texttt{cross,frag}}$; Eq. (5)) further boosts the performance by $74.7 \to 75.1$, capturing fine-grained semantics of molecules. Torsional angle prediction ($\mathcal{L}_{\texttt{tor}}$; Eq. (6)) further improves the performance by $75.1 \to 75.5$ by directly injecting the 3D local information into 2D-GNN. These results confirm that Holi-Mol effectively utilizes both 2D and 3D fragments for multi-view pretraining. Notably, ours with only single-view (2D) learning outperforms Mole-BERT (Xia et al., 2023), which is the prior state-of-the-art pretraining method on 2D molecular dataset. Detailed results can be found in Appendix I.

## 5 Related work

**Multi-view molecular representation learning**. Recent works have incorporated *multiple views* of a molecule (e.g., 2D topology and 3D geometry) into molecular representation learning (MRL) frameworks (Zhu et al., 2021a; Stärk et al., 2022; Liu et al., 2022b; Fang et al., 2022; Zhu et al., 2022; Luo et al., 2022). In particular, training 2D-GNNs with multi-view MRL has gained much attention to alleviate the cost to obtain 3D geometry of molecules (Stärk et al., 2022; Liu et al., 2022b). However, they focus on molecule-level objectives, which could lack capturing the local semantics (Yao et al., 2022). In this work, we develop a fragment-based multi-view MRL framework to incorporate local semantics.

**Single-view molecular representation learning**. One of the single-view (i.e., 2D topological or 3D geometric graph) molecular representation learning techniques is predictive pretext tasks. For example, those methods reconstruct the corrupted input as pre-defined pretext tasks (Hamilton et al., 2017; Rong et al., 2020a; Hu et al., 2020a; Zhang et al., 2021; Zhou et al., 2022; Jiao et al., 2022; Zaidi et al., 2022). Another large portion of technique is contrastive learning. For example, You et al. (2021) utilize augmentation schemes to produce a positive view of molecular graphs, and Fang et al. (2021); Sun et al. (2021); Wang et al. (2022) mitigate the effect of semantically similar molecules in the negative samples (Zhu et al., 2021b).

Recently, substructures of molecules has also been considered in molecular representation learning. For example, You et al. (2020); Wang et al. (2021); Zhang et al. (2020) construct a positive view of a molecule as its single substructure (i.e., subgraph) and Wang et al. (2022); Luong & Singh (2023) repels representations of

fragments from intra- and inter- molecular substructures. Compared to these works, our framework considers both global (molecule-level) and local features (substructure-level) of molecules in a unified framework.

## 6 Conclusion

We present Holi-Mol, a new multi-view molecular representation learning method to focus on both global (molecule-wise) and local (fragment-wise) information of molecules. With this insight, we propose a contrastive objective to regard the set of fragments as a positive view of a given molecule. Moreover, we introduce the fragment-based cross-view contrastive objective for 2D and 3D views of a molecule and the torsional angle prediction task to exploit 3D local information. Extensive experiments show that Holi-Mol outperforms prior methods in molecular property prediction tasks, thanks to our pretraining strategy guided by both molecule itself and its fragments.

## Broader Impact Statement

This work will facilitate research in molecular representation learning, which can speed up the processing of many important downstream tasks such as predicting side-effect of drugs. However, malicious use of well-learned molecular expressions poses a potential threat of creating hazardous substances, such as toxic chemical substances or biological weapons. On the other hand, molecular representation is also essential for creating defense mechanisms against harmful substances, so the careful use of our work, Holi-Mol, can lead to more positive effects.

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

# Supplementary Material
## Appendix: Holistic Molecular Representation Leraning
## via Multi-view Fragmentation

## A Experimental details

**Self-supervised pretraining details.** We follow the training setup considered in GraphMVP (Liu et al., 2022b): Specifically, we use a batch size of 256 and no weight decay. We use {`Nodedrop`, `Attrmask`, `identity`} randomly, i.e., $\frac{1}{3}$ probability for each fragment and the original 2D molecular graphs, and Gaussian noise $\mathcal{N}(0, I)$ to each coordinate of 3D molecular graphs. When `Nodedrop` or `Attrmask` is used, we drop/mask the portion of 0.1 vertices from the total vertices. For self-supervised pretraining, we train for 100 epochs using Adam optimizer (Kingma & Ba, 2014) with a learning rate of 0.001 and no dropout. For transfer learning to the QM9 (Ramakrishnan et al., 2014) dataset, we train with 310k entire unlabeled molecules from GEOM for 50 epochs. For semi-supervised leraning for the QM9 dataset, we train with 110k training molecules (without labels) from QM9 for 50 epochs. Our code is based on open-source codes of GraphMVP[4].

For Holi-Mol trained only with single view objective and other reproduced 2D baselines, we exclude explicit hydrogens in molecular graph, following the common frameworks of (You et al., 2020; 2021) for 2D molecular graphs. For Holi-Mol, 3D-InfoMax, GraphMVP, GraphMVP-C, and GraphMVP-G we include explicit hydrogens into molecular graph, following (Liu et al., 2022b) that utilizes the 3D coordinates of hydrogen atoms provided in GEOM dataset (Axelrod & Gomez-Bombarelli, 2022). For torsional angle prediction task, we use 2-layer MLP for $g_{\mathtt{tor}}$ and we construct the quartet of atoms $(s, u, v, t)$ for the fragmented bond $(u, v)$ so that $s, t$ are non-hydrogen atoms, and the binning of $y$ splits 0 to $2\pi$ into 18 uniform bins.

**Evaluation on MoleculeNet downstream tasks.** Following Liu et al. (2022b), we use *scaffold split* (Chen et al., 2012) in MoleculeNet experiments which splits the molecules based on their substructures. We use the split ratio train:validation:test = 80:10:10 for each downstream task dataset to evaluate the performance. For the consistency of the input graphs in pretraining and fine-tuning, we exclude implicit hydrogen atoms of molecules in fine-tuning dataset for single-view pretrained Holi-Mol and other reproduced 2D baselines and we include implicit hydrogen atoms of molecules in fine-tuning dataset for Holi-Mol, 3D-InfoMax, GraphMVP, GraphMVP-C, and GraphMVP-G. Experimental detail follows GraphMVP (Liu et al., 2022b); we fine-tune a pretrained 2D-GNN with an initialized linear layer for 100 epochs with Adam optimizer and a learning rate of 0.001, and dropout probability of 0.5. Our results are calculated by the test ROC-AUC score of the epoch with the best validation ROC-AUC score. Besides the ROC-AUC score of individual downstream tasks, we also report the average ROC-AUC score across downstream datasets.

**Evaluation on QM9 downstream tasks.** Following (Liu et al., 2021), we split the molecules in the QM9 (Ramakrishnan et al., 2014) dataset into 110,000 molecules for training, 10,000 molecules for validation, and 10,831 molecules for test. Our result is calculated by the test MAE score of the epoch with the best validation MAE score. We fine-tune a pretrained 2D-GNN with an initialized 2-layer multi layer perceptron for 1,000 epochs with Adam optimizer and StepLR scheduler with decay ratio of 0.5, and initial learning rate of 5e-4.

---

[4]`https://github.com/chao1224/GraphMVP`

## B   Baselines details

We compare our method with an extensive list of baseline methods in the literature of graph representation learning:

- *No pretraining* trains a model from scratch for downstream task.

- *AttrMask* (Hu et al., 2020a) learns representation by recovering the vertex features after masking them.

- *GPT-GNN* (Hu et al., 2020b) uses the graph generation task as a pretext task.

- *Infomax* (Sun et al., 2019) maximizes mutual information between global representations (i.e., graph representations) and local representations (i.e. path representation).

- *ContextPred* (Hu et al., 2020a) learns representation by predicting surrounding subgraph of specific node edge.

- *GraphLoG* (Xu et al., 2021) discriminates graph and subgraph pairs from their opposing pairs to preserve local similarity between various graphs, which leads to the embedding alignment of correlated graphs.

- *G-Contextual* (Rong et al., 2020a) learns representations by randomly masking local subgraphs of target nodes (or edges) and predicting these contextual properties from node embeddings.

- *G-Motif* (Rong et al., 2020a) predicts the occurrence of the semantic motifs extracted by using chemical prior.

- *GraphCL* (You et al., 2020) is a generic graph contrastive learning method based on their graph-agnostic augmentation schemes, which do not use any molecule-specific knowledge.

- *JOAO* (You et al., 2021) proposes min-max optimization processes to learn optimal data augmentation strategies dynamically from a pre-fixed candidate set of augmentations.

- *MGSSL* (Zhang et al., 2021) introduces a generative self-supervised objective to reconstruct a motif-tree.

- *MolCLR* (Wang et al., 2021) performs a contrastive learning with NT-Xent (Chen et al., 2020), constructing positive views of a molecule by proposed molecule augmentation schemes.

- *D-SLA* (Kim et al., 2022) extracts graph representations by learning the exact discrepancy between the original graph and the augmented graphs.

- *Mole-BERT* (Xia et al., 2023) utilizes VQ-VAE (Van Den Oord et al., 2017) to encode atoms as meaningful discrete values and then perform masked atoms modeling and triplet masked contrastive learning.

- *3D-InfoMax* (Stärk et al., 2022) proposes to consider 2D topological molecule graph and 3D geometric molecule graph from the same molecule as a positive view of each other.

- *GraphMVP, GraphMVP-G,* and *GraphMVP-C* (Liu et al., 2022b) regard 2D and 3D molecular graphs as a positive pair, and propose feature reconstruction of each view as a generative task.

## C Graph neural networks

**Graph Isomorphism Network (GIN).** We provide a detailed description of architecture of graph isomorphism network (GIN) (Xu et al., 2019), which we mainly consider as the feature extractor $f_{\text{2D}}(\cdot)$ in this paper. Particularly, GIN learns representation $\mathbf{h}_v^{(\ell)}$ by:

$$\mathbf{h}_v^{(\ell)} = \text{MLP}^{(\ell)}\big(\mathbf{h}_v^{(\ell-1)} + \sum_{u \in \mathcal{N}(v)} \big(\mathbf{h}_u^{(\ell-1)} + \mathbf{e}_{uv}^{(\ell-1)}\big)\big), \tag{8}$$

where $\mathbf{e}_{uv}^{(\ell-1)}$ is the embedding corresponding to the attribute of edge $\{u, v\} \in \mathcal{E}$.

**SchNet.** We consider SchNet (Schütt et al., 2017), which is a strong 3D graph neural network under fair comparison (Liu et al., 2022b) as our $f_{\text{3D}}(\cdot)$ in this paper. Particularly, SchNet learns representation $\mathbf{h}_v^{(\ell)} = \text{MLP}^{(\ell)}$ by:

$$\mathbf{h}_v^{(\ell)} = \text{MLP}^{(\ell)}\big(\sum_{u \in V} \big(\Phi(\mathbf{h}_u^{(\ell-1)}, \mathbf{r}_v, \mathbf{r}_u)\big)\big), \tag{9}$$

where $\Phi$ is the continuous-filter convolution layer and $\mathbf{r}_v$ is the 3D position of the vertex $v$.

# D Downstream dataset details

We perform transfer-learning on 8 benchmark binary classification datasets from MoleculeNet (Wu et al., 2018). More information on downstream tasks is described in Table 7.

- *BBBP* contains data on whether the compound is permeable to the blood-brain barrier.

- *Tox21* measures the toxicity of a compound and was used in the 2014 Tox21 Data Challenge.

- *ToxCast* includes multiple toxicity annotations of compounds collected after performing high-throughput screening tests.

- *Sider* refers to side effect resources, i.e., data on the marketed drugs and their side effects.

- *Clintox* is a dataset of comparison results between drugs approved through the FDA and drugs removed because of toxicity during clinical trials.

- *MUV* is a validation dataset of virtual screening technology. Specifically, it is subsampled in the PubChem BioAssay using refined nearest neighborhood analysis.

- *HIV* consists of data about capability to prevent HIV replication.

- *Bace* is collected dataset of compounds that could prevent (BACE-1).

Table 7: MoleculeNet downstream classification dataset statistics

| Dataset | BBBP | Tox21 | ToxCast | Sider | Clintox | MUV | HIV | Bace |
|---|---|---|---|---|---|---|---|---|
| Number of molecules | 2,039 | 7,831 | 8,575 | 1,427 | 1,478 | 93,087 | 41,127 | 1,513 |
| Number of tasks | 1 | 12 | 617 | 27 | 2 | 17 | 1 | 1 |
| Avg. Node | 24.06 | 18.57 | 18.78 | 33.64 | 26.15 | 24.23 | 25.51 | 34.08 |
| Avg. Degree | 51.90 | 38.58 | 38.52 | 70.71 | 55.76 | 52.55 | 54.93 | 73.71 |

We also perform transfer-learning on 12 benchmark regression tasks from QM9 (Ramakrishnan et al., 2014). More information on downstream tasks is described in Table 8.

Table 8: QM9 downstream regression tasks

| Task | Summary | Unit |
|---|---|---|
| ZPVE | Zero point vibrational energy | meV |
| $\mu$ | Dipole moment | $D$ |
| $\alpha$ | Isotropic polarizability | $a_0^3$ |
| $C_v$ | Heat capacity at $298.15K$ | $\mathrm{cal/mol \cdot K}$ |
| LUMO | Lowest unoccupied molecular orbital energy | meV |
| HOMO | Highest occupied molecular orbital energy | meV |
| $\varepsilon_{gap}$ | Gap between HOMO and LUMO | meV |
| $R^2$ | Electronic spatial extent | $a_0^2$ |
| $U_0$ | Internal energy at $0K$ | meV |
| $U_{298}$ | Internal energy at $0K$ | meV |
| $H_{298}$ | Enthalpy at $0K$ | meV |
| $G_{298}$ | Gibbs energy at $0K$ | meV |

# E    Detailed analysis of results on MoleculeNet

Table 9: Statistics of training and test samples of MoleculeNet dataset under scaffold split (Chen et al., 2012). The number of atoms are calculated in consideration of hydrogen atoms.

| Average # of atoms | BBBP | Tox21 | ToxCast | Sider | Clintox | MUV | HIV | Bace |
|---|---|---|---|---|---|---|---|---|
| Train | 43.6 | 33.0 | 33.0 | 57.8 | **49.6** | 42.6 | 45.3 | 63.5 |
| Test | 52.1 | 49.8 | 52.3 | 102.0 | **43.0** | 44.8 | 45.5 | 67.9 |

Table 10: Relative ROC-AUC improvement to GraphMVP-G (the best performing baseline). We compare the results on the full test set (in Table 1) and the subset of the test set which contains the molecules with less atoms than the average number of atoms in training molecules.

| Improvement of Holi-Mol from GraphMVP-G | BBBP | Bace | MUV | HIV |
|---|---|---|---|---|
| Full test molecules | 1.3 | 1.0 | -0.1 | 1.0 |
| Test molecules with less atoms than # Avg. atoms in training set | **2.3** | **1.1** | **2.9** | **1.5** |

In Table 1, the improvement of Holi-Mol is most dramatic on the Clintox dataset. We observe that Clintox is unique in that it has a higher average number of atoms in the training set compared to the test set (see Table 9). Furthermore, we observe that the performance gap between Holi-Mol and GraphMVP-G (the strongest baseline) tends to widen as the number of atoms in test molecules decreases. To illustrate this point, Table 10 compares the performance gap when evaluated on the full test set and on the molecules in the test set with fewer atoms than the average in the training set (we consider the downstream tasks with |Avg. # of atoms in the training set molecules - Avg. # of atoms in the test set molecules| < 10, due to the stability of evaluation). Holi-Mol's enhanced performance on smaller molecules can be explained by its capacity to learn fine-grained molecular features through fragmentation.

# F    Complexity analysis

Table 11: Analysis of the training time of each method. We compare the most competitive baselines in Table 1. The training time is calculated based on the same setup of Table 1 with a single NVIDIA GeForce RTX 3090 GPU.

| | MGSSL | Mole-BERT | 3D-Infomax | GraphMVP-G | Holi-Mol (Ours) |
|---|---|---|---|---|---|
| Training time (hours) | 17.5 | 0.7 | 0.9 | 4.5 | 1.9 |
| Test ROC-AUC score | 70.8 | 71.8 | 73.4 | 74.1 | **75.5** |

In Table 11, we provide the running time analysis of each method and the corresponding test ROC-AUC score on the MoleculeNet benchmark. We observe that MGSSL (Zhang et al., 2021) requires extensive training cost because of the auto-regressive predictive pretext task. GraphMVP-G (Liu et al., 2022b) utilizes several 3D conformer structures, leading to higher training complexity. Overall, our Holi-Mol significantly improves the prior best-performing baseline, GraphMVP-G (Liu et al., 2022b), in the test ROC-AUC score (74.1 → 75.5; higher is better) with less training time (4.5 → 1.9; lower is better).

# G   Detailed results on semi-supervised learning

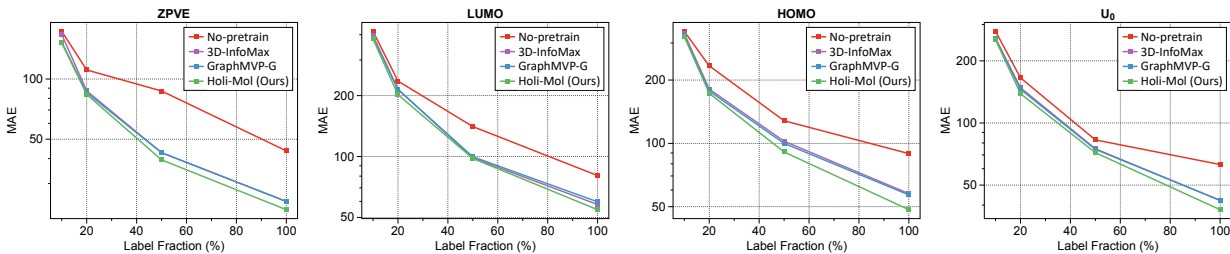

Figure 2:   Comparison of test MAE score of semi-supervised learning with different fraction of labeled samples on QM9 downstream quantum property regression benchmarks. We pretrain GIN (Xu et al., 2019) as the 2D-GNN architecture with 110k molecules from QM9 pretraining dataset.

Table 12: Comparison of test MAE score of semi-supervised learning on the QM9 downstream quantum property regression benchmarks. We pretrain GIN (Xu et al., 2019) as the 2D-GNN architecture with 110k QM9 training set and fine-tune on 10% subset of QM9 training set. We mark the best score bold.

| Methods | ZPVE $\downarrow$ | $\mu \downarrow$ | $\alpha \downarrow$ | $C_v \downarrow$ | LUMO $\downarrow$ | HOMO $\downarrow$ | $\varepsilon_{gap} \downarrow$ | $R^2 \downarrow$ | $U_0 \downarrow$ | $U_{298} \downarrow$ | $H_{298} \downarrow$ | $G_{298} \downarrow$ |
|---|---|---|---|---|---|---|---|---|---|---|---|---|
| - | 173.1 | 0.339 | 2.67 | 0.882 | 415.5 | 340.7 | 680.8 | 20.6 | 278.0 | 301.3 | 299.9 | 274.1 |
| Mole-BERT Xia et al. (2023) | 163.6 | 0.330 | 2.65 | 0.877 | 402.2 | 329.0 | 671.2 | 20.2 | 266.6 | 287.1 | 285.3 | 250.9 |
| Pretrained on 110k 2D and 3D molecular graphs of QM9 and fine-tuned on 10% 2D molecular graphs of QM9 | | | | | | | | | | | | |
| 3D-Infomax Stärk et al. (2022) | 166.7 | 0.325 | 2.59 | 0.878 | 395.3 | 332.7 | 672.7 | 20.4 | 257.5 | 284.1 | 283.9 | 249.4 |
| GraphMVP-G Liu et al. (2022b) | 152.6 | 0.324 | 2.58 | 0.872 | 388.3 | 325.8 | 662.7 | 19.9 | 255.4 | 281.4 | 271.7 | 245.3 |
| **Holi-Mol (Ours)** | **151.5** | **0.322** | **2.51** | **0.869** | **381.0** | **321.2** | **650.5** | **19.8** | **252.9** | **279.4** | **269.1** | **243.6** |

Table 13: Comparison of test MAE score on QM9 downstream quantum property regression benchmarks when additional 3D geometric is available in fine-tuning. We utilize GIN (Xu et al., 2019) and SchNet Schütt et al. (2017) for 2D and 3D encoder, respectively. We mark the best score bold.

| Methods | ZPVE $\downarrow$ | LUMO $\downarrow$ | HOMO $\downarrow$ | $U_0 \downarrow$ |
|---|---|---|---|---|
| - | 1.67 | 26.8 | 32.9 | 14.5 |
| GraphMVP-G Liu et al. (2022b) | 1.65 | 26.1 | 32.8 | 14.3 |
| GeoSSL-DDM Liu et al. (2022a) | 1.63 | **25.8** | 32.0 | 14.1 |
| 3D-EMGP Jiao et al. (2022) | 1.63 | 25.9 | 31.9 | 14.0 |
| Holi-Mol (Ours) | **1.61** | **25.8** | **31.6** | **13.9** |

In this section, we provided detailed results for semi-supervisd learning on the QM9 (Ramakrishnan et al., 2014) dataset. Figure 2 shows the test MAE score across different label fractions after pretrained with the QM9 training dataset. We choose 4 downstream tasks which yields the highest performance gap after pretraining compared to non-pretraining (we exclude $\varepsilon_{gap} := |\text{HOMO} - \text{LUMO}|$ since we already include HOMO and LUMO). As visualized, Holi-Mol consistently outperforms the considered baselines. Table 12 shows the results for all 12 downstream tasks of QM9 when fine-tuned with 10% of training data. For all downstream tasks, Holi-Mol achieves the best performance.

Although our primary focus is to pretrain 2D-GNN (Stärk et al., 2022; Liu et al., 2022b), we explore the possibility of Holi-Mol in the scenario when 3D information is additionally available in the fine-tuning phase. We follow the same experimental setup as detailed in Table 2 (i.e., pretraining on QM9 and using SchNet as the 3D GNN). We test 4 downstream tasks (ZPVE, LUMO, HOMO, and $U_0$) in the QM9 dataset following the setup in Table 3 and Figure 2. We additionally compare with recent 3D geometric pretraining baselines, 3D-EMGP (Jiao et al., 2022) and GeoSSL-DDM (Liu et al., 2022a). For Holi-Mol and GraphMVP-G, we consider fine-tuning 2D and 3D encoders jointly. Here, we note that one of the benefits of Holi-Mol compared to geometric pretraining baselines (with only 3D encoder) is that we can utilize both 2D and 3D encoders. In Table 13, we observe that Holi-Mol still shows reasonable performance when 3D information is additioanlly available in fine-tuning dataset.

## H   Ablation on fragment-based positive view construction of Holi-Mol

Table 14: Comparison of positive view construction strategies for multi-view molecular contrastive learning framework. We report the test ROC-AUC score on the MoleculeNet downstream property classification benchmarks. We pretrain GIN (Xu et al., 2019) as the 2D-GNN architecture with 50k molecules from the GEOM dataset (Axelrod & Gomez-Bombarelli, 2022), following Liu et al. (2022b). We report mean and standard deviation over 3 different seeds. We bold the best average score.

| Positive view construction | Fragmentation strategy | BBBP | Tox21 | ToxCast | Sider | Clintox | MUV | HIV | Bace | Avg. |
|---|---|---|---|---|---|---|---|---|---|---|
| Nodedrop, Subgraph | - | $69.3_{\pm1.4}$ | $75.0_{\pm0.4}$ | $63.7_{\pm0.4}$ | $60.4_{\pm1.4}$ | $88.3_{\pm0.6}$ | $76.2_{\pm1.9}$ | $76.2_{\pm1.5}$ | $78.3_{\pm0.4}$ | 73.4 |
| **A set of fragments (Ours)** | Random bond deletion | $69.3_{\pm1.0}$ | $73.8_{\pm0.9}$ | $63.9_{\pm0.5}$ | $59.9_{\pm1.2}$ | $91.4_{\pm2.3}$ | $76.8_{\pm0.7}$ | $74.6_{\pm3.1}$ | $78.3_{\pm2.5}$ | 73.5 |
| | Random non-ring bond deletion | $69.5_{\pm0.9}$ | $73.7_{\pm0.2}$ | $64.0_{\pm0.1}$ | $60.5_{\pm0.5}$ | $93.2_{\pm1.5}$ | $77.3_{\pm2.5}$ | $75.2_{\pm0.9}$ | $78.8_{\pm0.4}$ | 74.0 |
| | **BRICS decomposition (Ours)** | $\mathbf{71.4}_{\pm0.4}$ | $\mathbf{75.2}_{\pm0.7}$ | $\mathbf{65.1}_{\pm0.8}$ | $\mathbf{61.0}_{\pm0.6}$ | $\mathbf{95.2}_{\pm1.0}$ | $\mathbf{77.6}_{\pm1.0}$ | $\mathbf{76.3}_{\pm0.4}$ | $\mathbf{82.3}_{\pm1.6}$ | **75.5** |

In Table 14, we provide a full result of Table 5 in Section 4.3. We conduct an ablation study on regarding the set of fragments as a positive view of a molecule. Again, we emphasize that the result implies that considering chemically informative structures is a key component of Holi-Mol.

## I   Component ablation of Holi-Mol

Table 15: Ablation of components for multi-view molecular contrastive learning framework. We report the test ROC-AUC score on the MoleculeNet downstream property classification benchmarks. We pretrain GIN (Xu et al., 2019) as the 2D-GNN architecture with 50k molecules from the GEOM dataset (Axelrod & Gomez-Bombarelli, 2022), following Liu et al. (2022b). We report mean and standard deviation over 3 different seeds. We mark the best mean score to be bold.

| Pretraining data | Multi-view interaction | | | BBBP | Tox21 | ToxCast | Sider | Clintox | MUV | HIV | Bace | Avg. |
|---|---|---|---|---|---|---|---|---|---|---|---|---|
| | Molecule-level | Fragment-level | Torsion-level | | | | | | | | | |
| Single-view (2D) | - | - | - | $71.0_{\pm0.3}$ | $75.3_{\pm0.8}$ | $62.8_{\pm0.4}$ | $60.3_{\pm1.1}$ | $79.1_{\pm2.2}$ | $74.1_{\pm0.5}$ | $75.9_{\pm1.2}$ | $80.7_{\pm1.3}$ | 72.4 |
| Multi-view (2D & 3D) | ✓ | - | - | $68.2_{\pm0.6}$ | $\mathbf{75.6}_{\pm1.5}$ | $64.6_{\pm0.2}$ | $60.8_{\pm0.8}$ | $94.9_{\pm0.8}$ | $\mathbf{77.7}_{\pm1.2}$ | $\mathbf{76.3}_{\pm0.5}$ | $79.5_{\pm0.3}$ | 74.7 |
| | ✓ | ✓ | - | $71.0_{\pm0.8}$ | $75.3_{\pm0.9}$ | $64.4_{\pm0.3}$ | $\mathbf{61.6}_{\pm2.6}$ | $95.1_{\pm1.5}$ | $76.4_{\pm1.6}$ | $76.2_{\pm0.7}$ | $80.9_{\pm2.6}$ | 75.1 |
| | ✓ | ✓ | ✓ | $\mathbf{71.4}_{\pm0.4}$ | $75.2_{\pm0.7}$ | $\mathbf{65.1}_{\pm0.8}$ | $61.0_{\pm0.6}$ | $\mathbf{95.2}_{\pm1.0}$ | $77.6_{\pm1.0}$ | $\mathbf{76.3}_{\pm0.4}$ | $\mathbf{82.3}_{\pm1.6}$ | **75.5** |

In Table 15, we provide a full result of Table 6 in Section 4.3. We validate that each components of Holi-Mol has an individual effect in improving the performance of multi-view pretraining.

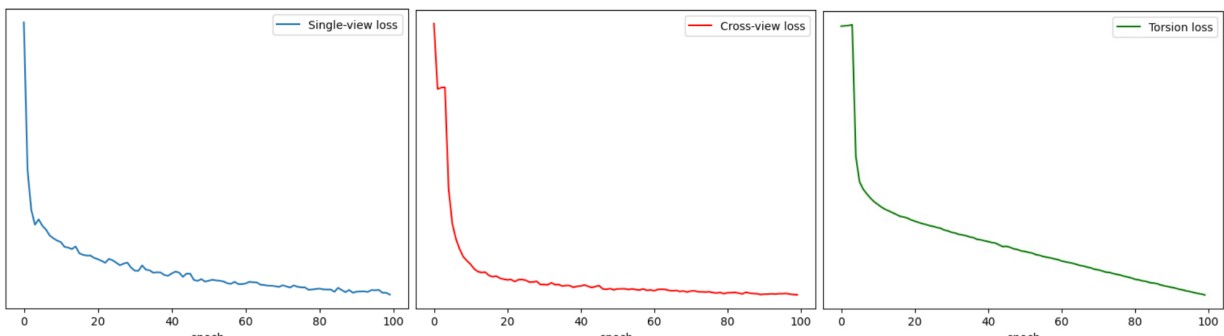

Figure 3:   Training curves of each component: (1) Single-view objective $\mathcal{L}_{\texttt{single}}$, (2) Cross-view objective $\mathcal{L}_{\texttt{cross}}$, and (3) Torsion objective $\mathcal{L}_{\texttt{tor}}$.

In Figure 3, we report the training curves of our objectives. Our training objectives carefully designed to learn crucial but non-overlapping features of unlabeled molecules, e.g., fragment/molecule-level or 2D/3D view. As a result, the individual losses consistently decrease during training.

## J  Case study

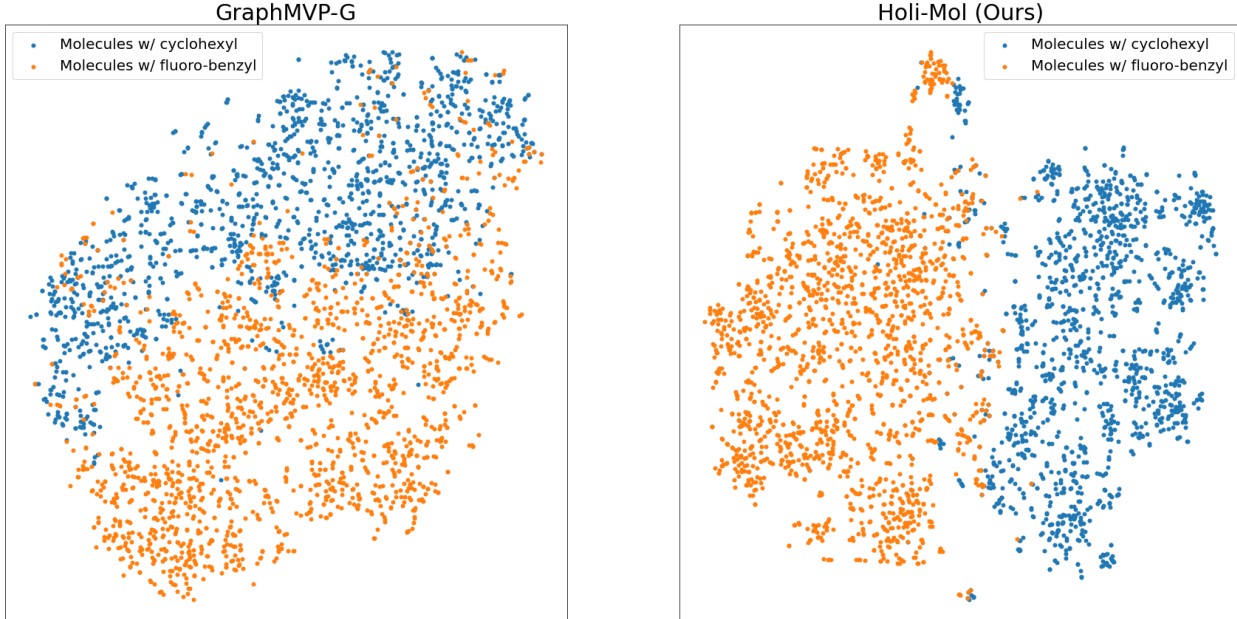

Figure 4:  t-SNE plot of the representations obtained by GraphMVP-G (left) and Holi-Mol (right).

The global properties, i.e., labels, of a molecule are closely associated with its meaningful local substructures, i.e., fragments. For example, the ratio of 1-labeled molecules in the BACE (0-1 labeled) dataset is 45%. However, the ratio drops to 29% for samples with cyclohexyl group, and the ratio goes up to 60% for samples with fluoro-benzyl group. These variations can be attributed to the hydrophobicity of the cyclohexyl group and the aromaticity or hydrophilicity of the fluoro-benzyl group. However, the current state-of-the-art methods, e.g., GraphMVP-G, do not take into account the relationship between the local and global information of molecules. Thus, our method aims to capture such a relationship in molecular representation learning. For empirical support, in Figure 4, we provide t-SNE of the molecular representations containing the discussed semantic-determining groups (cyclohexyl and fluoro-benzyl) in the GEOM pre-training dataset. Holi-mol shows superior discriminability for such groups, which supports the overall improvements in downstream tasks in Table 1,2, and 3. This validates the efficacy of our novel approach, which captures the interaction between a molecule and its constituent fragments within a unified framework.

## K  Results on PCQM4Mv2

Table 16: Results on the PCQM4Mv2 dataset.

| Methods | - | 3D-Infomax | GraphMVP-G | **Holi-Mol (Ours)** |
|---|---|---|---|---|
| Validation MAE | 0.1703 | 0.1692 | 0.1693 | **0.1688** |

In Table 16, we verify the effectiveness of our method in the PCQM4Mv2 dataset. Specifically, we fine-tune the pre-trained models of each baseline with a small fraction of the training data, i.e., 10%, to follow the conventional evaluation setup of molecular representation learning where the labeled data is fewer than unlabeled data. Our method achieves superior performance in terms of validation MAE.

## L    Comparison with fingerprints

Table 17: Comparison with fingerprint representation (Morgan, 1965) on MoleculeNet.

| Representations | BBBP | Tox21 | ToxCast | Sider | Clintox | MUV | HIV | Bace | Avg. |
|---|---|---|---|---|---|---|---|---|---|
| Morgan fingerprints (Morgan, 1965) | $67.5_{\pm 1.5}$ | $70.4_{\pm 0.6}$ | $56.7_{\pm 0.1}$ | $57.7_{\pm 0.5}$ | $75.4_{\pm 2.1}$ | $66.4_{\pm 0.2}$ | $66.1_{\pm 0.2}$ | $76.4_{\pm 1.2}$ | 67.1 |
| **Ours** | $\mathbf{71.4}_{\pm 0.4}$ | $\mathbf{75.2}_{\pm 0.7}$ | $\mathbf{65.1}_{\pm 0.8}$ | $\mathbf{61.0}_{\pm 0.6}$ | $\mathbf{95.2}_{\pm 1.0}$ | $\mathbf{77.6}_{\pm 1.0}$ | $\mathbf{76.3}_{\pm 0.4}$ | $\mathbf{82.3}_{\pm 1.6}$ | **75.5** |

In Table 17, we compare the representations obtained from our method with the commonly used Morgan fingerprints (Morgan, 1965) of molecules and fragments, where the attention pooling is used to aggregate the fingerprints of fragments. The results show that our learned $r_i$ and $r_i^j$ consistently outperform the fingerprints, which verifies the effectiveness our learned representations.

## M    Ananlysis on fine-tuning

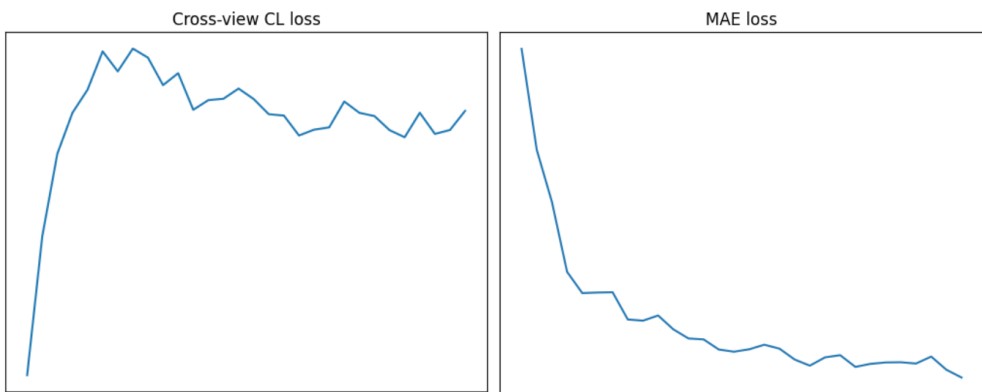

Figure 5:    Plot of the cross-view objective $\mathcal{L}_{\mathtt{cross}}$ and the MAE objecive in fine-tuning. Note that the objective function in fine-tuning is the MAE objective to the target label (not the cross-view objective).

In Figure 5, we provide an analysis on fine-tuning. We pre-train with a half of the QM9 dataset and then fine-tune exclusively on the 2D view of another half of the QM9 dataset (which has target labels). In the fine-tuning phase, the MAE improves while the contrastive loss between the 2D and 3D views increases, which is natural since we do not impose the contrastive objective in fine-tuning (rather we apply the supervised objective to learn the labels). In other words, our pre-training scheme learns a "good initialization" of a GNN using both 2D and 3D views and then we additionally inject the target-specific 2D information of molecules using the learned GNN via fine-tuning.

## N    Variance of QM9 experiment

Table 18: Standard deviation of the experiment in Table 2. The results are based on 3 seeds.

| Methods | ZPVE ↓ | LUMO ↓ | HOMO ↓ | $U_0$ ↓ |
|---|---|---|---|---|
| - | 43.7 | 80.5 | 89.4 | 62.9 |
| 3D-Infomax Stärk et al. (2022) | 27.0 | 63.4 | 55.2 | **38.8** |
| GraphMVP-G Liu et al. (2022b) | **24.1** | 59.1 | 53.8 | **39.9** |
| Holi-Mol (Ours) | $\mathbf{24.0}_{\pm 0.1}$ | $\mathbf{57.2}_{\pm 0.3}$ | $\mathbf{51.8}_{\pm 0.3}$ | $39.0_{\pm 1.3}$ |

# O   Discussion on evaluation setup

Table 19:   Test MAE score on the QM9 downstream quantum property regression benchmarks. We mark the best score bold. The contents are the same as Table 2.

| Methods | ZPVE ↓ | $\mu$ ↓ | $\alpha$ ↓ | $C_v$ ↓ | LUMO ↓ | HOMO ↓ | $\varepsilon_{gap}$ ↓ | $R^2$ ↓ | $U_0$ ↓ | $U_{298}$ ↓ | $H_{298}$ ↓ | $G_{298}$ ↓ |
|---|---|---|---|---|---|---|---|---|---|---|---|---|
| - | 43.7 | 0.059 | 0.400 | 0.144 | 80.5 | 89.4 | 171.0 | 3.27 | 62.9 | 61.8 | 57.0 | 48.1 |
| Pretrained on 310k 2D molecular graphs of GEOM and fine-tuned on 2D molecular graphs of QM9 | | | | | | | | | | | | |
| 3D-Infomax Stärk et al. (2022) | 27.0 | 0.051 | 0.355 | 0.126 | 63.4 | 55.2 | 103.8 | 2.99 | **38.8** | 45.6 | 41.0 | 40.8 |
| GraphMVP-G Liu et al. (2022b) | 24.1 | 0.051 | 0.367 | 0.123 | 59.1 | 53.8 | 100.4 | 2.97 | 39.9 | 44.2 | 41.0 | 40.3 |
| **Holi-Mol (Ours)** | **24.0** | **0.049** | **0.353** | **0.121** | **57.1** | **51.8** | **97.1** | **2.90** | 39.2 | **42.9** | **40.3** | **40.0** |

Table 20:   Test ROC-AUC score on the MoleculeNet molecular property classification benchmarks. We report mean and standard deviation over 3 different seeds. We mark the best mean score to be bold.

| Methods | BBBP | Tox21 | ToxCast | Sider | Clintox | MUV | HIV | Bace | Avg. |
|---|---|---|---|---|---|---|---|---|---|
| - | $65.4_{\pm2.4}$ | $74.9_{\pm0.8}$ | $61.6_{\pm1.2}$ | $58.0_{\pm2.4}$ | $58.8_{\pm5.5}$ | $71.0_{\pm2.5}$ | $75.3_{\pm0.5}$ | $72.6_{\pm4.9}$ | 67.2 |
| Pretrained with 310k 2D and 3D molecular graphs of GEOM and fine-tuned on 2D molecular graphs of MoleculeNet | | | | | | | | | |
| 3D-InfoMax (Stärk et al., 2022) | $68.8_{\pm0.2}$ | $75.0_{\pm1.5}$ | $64.4_{\pm0.4}$ | $61.7_{\pm0.2}$ | $89.7_{\pm1.3}$ | $76.9_{\pm0.8}$ | $75.3_{\pm0.5}$ | $79.8_{\pm0.6}$ | 73.9 |
| GraphMVP-G† (Liu et al., 2022b) | $69.8_{\pm0.8}$ | $75.5_{\pm0.1}$ | $64.1_{\pm0.2}$ | $61.1_{\pm0.7}$ | $90.1_{\pm0.9}$ | $77.6_{\pm0.5}$ | $75.7_{\pm0.5}$ | $80.3_{\pm0.7}$ | 74.3 |
| **Holi-Mol (Ours)** | $\mathbf{71.2}_{\pm0.8}$ | $\mathbf{75.7}_{\pm0.3}$ | $\mathbf{64.9}_{\pm0.4}$ | $\mathbf{61.7}_{\pm0.2}$ | $\mathbf{95.3}_{\pm0.2}$ | $\mathbf{77.6}_{\pm1.4}$ | $\mathbf{76.5}_{\pm0.4}$ | $\mathbf{82.4}_{\pm0.5}$ | **75.7** |

In Table 1, 2, and 3, we choose different pre-training setups to consider various practical scenarios. We first remark that the molecule space is very large. Thus, the fine-tuning distribution may often be quite different from the pre-training distribution. Therefore, we divide the evaluation setup into two cases based on the pre-training and fine-tuning distributions.

- In Table 1 and 2, we assume the pre-training and fine-tuning distributions are different. Specifically, we pre-train on the GEOM dataset and then fine-tune on the MoleculeNet (Table 1) and QM9 (Table 2), respectively.

- In Table 3, we assume the pre-training and fine-tuning distributions are the same. Specifically, we use the QM9 dataset for both pre-training and fine-tuning.

We also note that we use 50k molecules from the GEOM dataset in Table 1 for a fair comparison with existing works reported in Liu et al. (2022b). However, we find that there is no specific reason not to use the full GEOM dataset (310k molecules) for pre-training. Thus, in the evaluation on QM9 (Table 2), we use the entire 310k molecules of the GEOM dataset for pre-training the models.

Nevertheless, it is possible to pre-train a single model for all the downstream tasks. We further report the results on the downstream tasks using a single pre-trained model. Specifically, we use the pre-trained model based on the 310k molecules from the GEOM dataset (which is the setup of Table 2) for each method. Table 19 and 20 show that the single pre-trained model also outperforms the baselines on the downstream tasks (MoleculeNet and QM9).

