# OpenReview forum: "Holistic Molecular Representation Learning via Multi-view Fragmentation"
_TMLR — Accepted by TMLR_

### Review · Reviewer_wGgA · 2024-02-12

**Summary Of Contributions:**

The paper introduces an innovative framework that utilizes a multi-view fragmentation approach for the pre-training of molecules, considering both holistic and fragmented representations. The concept of employing both 2D and 3D perspectives of molecules presents a novel approach. To understand the interactions between molecular fragments, the authors ingeniously devise a torsional angle prediction method, facilitating the learning of correlations among fragments.

**Audience:**

Yes

**Claims And Evidence:**

Yes

**Requested Changes:**

Based on the identified weaknesses and questions:

1. **2D GNN Necessity:** Conduct an ablation study by substituting the 2D GNN embeddings for molecules and fragment sets with fingerprints to evaluate the critical role of 2D GNNs.

2. **Loss Sensitivity:** Plot the training loss for all three types of losses to determine if any of the losses are, in fact, increasing during training.

3. **Fine-Tuning Forgetting Issue:** Divide the GEOM dataset into two parts. Pre-train the model using both 2D and 3D views of the first part, then fine-tune it exclusively on the 2D view of the second part. Evaluate the contrastive loss between 2D and 3D views of the second part to investigate any increase in loss.

Minor:

4. **Standard Deviation Reporting:** It is advisable to include the standard deviation (STD) in Table 2, as the improvements depicted appear to be marginal. This additional data would provide a clearer understanding of the results' variability and significance.

**Strengths And Weaknesses:**

Strengths:

1. The paper is novel in its approach to utilizing both holistic and segmented representations of molecules.

2. It incorporates the use of both 2D and 3D perspectives for the self-training of molecules.

3. It introduces the use of torsional angle prediction as a method to understand the relationships between molecular fragments.


Weaknesses/Questions:

1. **The Necessity of 2D GNN:** Is the application of a 2D GNN for molecule embedding mandatory? Could molecular fingerprints serve as an alternative to ￥r_i$  and $r_i^j$s?

2. **Trend of Multimodal Large Models:** Given the current interest in multimodal large models, and considering multi-view as a subset of multimodal, why opt for contrastive learning across views instead of employing cross-attention mechanisms like those in CLIP?

3. **Implementation of 3D GNN:** In employing SchNet for 3D positions, were the molecular fragments re-centered?

4. **Loss Sensitivity:** With the model incorporating multiple types of losses, is there a risk of these losses competing against each other?

5. **Fine-Tuning Forgetting Issue:** When the model is pre-trained on both 2D and 3D molecular graphs but fine-tuned solely on 2D graphs, would this enlarge the contrastive loss between the 2D and 3D representations?

---

> ### Author Response · Authors · 2024-02-29
> **Response to Reviewer wGgA (1/2)**
>
> Dear reviewer wGgA,
>
>
> We sincerely appreciate your efforts in reviewing our manuscript. We respond to each comment in the following content. In the revised manuscript, we have marked the revisions with “blue”.
>
> ------
> **[Q1/C1] The Necessity of 2D GNN: Is the application of 2D GNN for molecule embedding mandatory? Could molecular fingerprints serve as an alternative?**
>
>
> The application of 2D GNN for molecule embedding is mandatory to achieve best performance and molecular fingerprints cannot serve as an alternative to $r_i$ and $r_i^j$ in our representation learning framework. We remark that we aim to learn chemically meaningful $r_i$ and $r_i^j$ via 2D GNN. Therefore, fingerprints cannot replace $r_i$ and $r_i^j$ since they are fixed values for given molecules or fragments, i.e., they are not learnable. We emphasize that the learning framework allows us to enrich the molecule representations ($r_i$ and $r_i^j$) by considering various aspects of molecules, i.e., molecule-level and fragment-level features. For your information, we conduct an experiment to show the superiority of our learned 2D GNN representations over fingerprints. Specifically, we train a classifier with Morgan fingerprints [1] without representation learning. The results show that our learned representations consistently outperform the fingerprints, which validates our framework with the application of 2D GNN. We added the results in Appendix L of our revised manuscript.
>
> \begin{array}{l|cccccccc|c}
> \hline
> & \text{BBBP} & \text{Tox21} & \text{ToxCast} & \text{Sider} & \text{Clintox} & \text{MUV} & \text{HIV} & \text{Bace} & \text{Avg.}\newline
> \hline
> \text{Fingerprints} & 67.5\pm{1.5} & 70.4\pm{0.6} & 56.7\pm{0.1} & 57.7\pm{0.5} & 75.4\pm{2.1} & 66.4\pm{0.2} & 66.1\pm{0.2} & 76.4\pm{1.2} & 67.1 \newline
> \hline
> \text{Ours} & \textbf{71.4}\pm{0.4} & \textbf{75.2}\pm{0.7} & \textbf{65.1}\pm{0.8} & \textbf{61.0}\pm{0.6} & \textbf{95.2}\pm{1.0} & \textbf{77.6}\pm{1.0} & \textbf{76.3}\pm{0.4} & \textbf{82.3}\pm{1.6} & \textbf{75.5} \newline
> \hline
> \end{array}
>
> [1] Morgan and Harry, The Generation of a Unique Machine Description for Chemical Structures - a Technique Developed at Chemical Abstracts Service, Journal of Chemical Documentation 1965
>
> ------
> **[Q2] Trend of Multimodal Large Models: Why opt for contrastive learning across views instead of employing cross-attention like CLIP?**
>
> We opt for contrastive learning across views with 2D/3D GNNs to mitigate the lack of 3D information in real-world fine-tuning scenarios, where such a setup has been known to be crucial in previous works [1,2]. Nevertheless, our concept of ``holistic’’ molecular representation learning via molecule-level and fragment-level features can also be applied to such multi-modal large molecule models (if exist) and such a direction would be a crucial and interesting future work.
>
> [1] Stark et al., 3D Infomax improves GNNs for Molecular Property Prediction, ICML 2022\
> [2] Liu et al., Pre-training Molecular Graph Representation with 3D Geometry, ICLR 2022
>
> -----
> **[Q3] Implementation of 3D GNN: In employing SchNet for 3D positions, were the molecular fragments re-centered?**
>
> We do not re-center the molecular fragments in employing SchNet for 3D positions. Indeed, re-centering the 3D fragments does not affect the representation of the 3D fragments since SchNet (as well as many other 3D GNNs) is a translation-invariant architecture.
>
>
> -----
> **[Q4/C2] Loss Sensitivity: With the model incorporating multiple types of losses, is there a risk of these losses competing against each other?**
>
>
> Our individual losses do not compete against each other. Specifically, each loss is carefully designed to learn crucial but non-overlapping features of molecules, e.g., fragment/molecule-level or 2D/3D view. As a result, the individual losses consistently decrease during training. We have included the training curves in Appendix M of the revised manuscript.

---

> ### Author Response · Authors · 2024-02-29
> **Response to Reviewer wGgA (2/2)**
>
> **[C4] Standard deviation in the QM9 experiment.**
>
> We appreciate your constructive suggestion. Following your suggestion, we provide the standard deviation of the results in Table 2 in the table below. We added the results in Appendix O of the revised manuscript.
>
> \begin{array}{l|cccc}
> \hline
> & \text{ZPVE} & \text{LUMO} & \text{HOMO} & U_0 \newline
> \hline
> \text{-} & 43.7 & 80.5 & 89.4 & 62.9 \newline
> \text{3D-Infomax} & 27.0 & 63.4 & 55.2 & \textbf{38.8} \newline
> \text{GraphMVP-G} & \textbf{24.1} & 59.1 & 53.8 & \textbf{39.9} \newline
> \hline
> \text{Holi-Mol (Ours)} & \textbf{24.0}\pm{0.1} & \textbf{57.2}\pm{0.3} & \textbf{51.8}\pm{0.3} & \textbf{39.0}\pm{1.3}  \newline
> \hline
> \end{array}

---

> > ### Comment · Reviewer_wGgA · 2024-03-06
> > **Concerns about Question 5**
> >
> > Could you provide more experiments for the Q5/C3?

---

> ### Author Response · Authors · 2024-03-06
> **Response to Reviewer wGgA**
>
> Dear reviewer wGgA, we sincerely appreciate your comment. We included this discussion in Appendix N of the revised manuscript.
>
> ----
> **[Q5/C3] Fine-tuning forgetting issue when fine-tuning solely on 2D graphs**
>
> We first note that there is no forgetting issue of 3D information in fine-tuning on 2D graphs. If this is the case, pre-training on 2D graphs and 2D/3D graphs should show similar performance after fine-tuning on 2D graphs. However, pre-training on 2D/3D graphs significantly outperforms the case of pre-training solely on 2D graphs (72.4 $\rightarrow$ 75.5 in MoleculeNet, Table 15).
>
> We also clarify our pre-training and fine-tuning setup. We pre-train on 2D/3D unlabeled molecular graphs (the GEOM dataset) with our proposed self-supervised learning objective (in Eq. (7)) which includes 2D/3D contrastive loss. Then, we fine-tune the learned 2D GNN and additional prediction head on 2D labeled molecular graphs (the MoleculeNet benchmark) with supervised loss. Thus, it is not possible to ``fine-tune exclusively on the 2D view of the second part of the GEOM dataset’’ since there are no labels to apply supervised loss in the GEOM dataset.
>
> Nevertheless, for your information, we pre-train with a half of the QM9 dataset and then fine-tune exclusively on the 2D view of another half of the QM9 dataset (which has target labels). In the fine-tuning phase, the MAE improves while the contrastive loss between the 2D and 3D views increases, which is natural since we do not impose the contrastive objective in fine-tuning (rather we apply the supervised objective to learn the labels). In other words, our pre-training scheme learns a ``good initialization’’ of a GNN using both 2D and 3D views and then we additionally inject the target-specific 2D information of molecules using the learned GNN via fine-tuning. We provide training and accuracy curves in the revised manuscript.

---

### Review · Reviewer_iXqL · 2024-02-18

**Summary Of Contributions:**

This paper proposes a MRL model that decomposes a molecule into a set of fragments, to associate a global graph structure to a set of local substructures, and also consider the 3D geometry of molecules as another view for contrastive learning.

**Audience:**

Yes

**Claims And Evidence:**

No

**Requested Changes:**

Referring to Weaknesses.

**Strengths And Weaknesses:**

Strengths:

- The manuscript is well-written and easy to follow.
- The proposed method is simple and effective.

Weaknesses:

- The contribution of the proposed idea appears to be relatively incremental, merging concepts of 2D & 3D and fragment-based contrastive learning, both of which have been extensively explored in prior research. This raises questions about the significant advancement over existing methodologies.

- The experimental design is somewhat confusing. The inclusion of three distinct training configurations (50k, 310k, and 110k datasets) in Tables 1, 2, and 3 lacks a clear justification. In MRL, the goal is usually to develop a single pre-trained model capable of addressing a variety of downstream tasks, rather than creating bespoke models for individual tasks.
- It remains uncertain whether the proposed method can be applied to tasks requiring 3D inputs, as the paper currently only demonstrates its applicability to 2D input tasks.
- Providing results on the PCQM4MV2 dataset, particularly on the test set, would offer a more comprehensive assessment of the model's effectiveness and applicability.

---

> ### Author Response · Authors · 2024-02-29
> **Response to Reviewer iXqL**
>
> Dear reviewer iXqL,
>
>
> We sincerely appreciate your efforts in reviewing our manuscript. We respond to each comment in the following content. In the revised manuscript, we have marked the revisions with “blue”.
>
> ------
> **[W1] Fragment-based contrastive learning and merging 2D&3D have been explored.**
>
> We politely disagree with the reviewer’s opinion. How we utilize the concept of “fragmentation” is completely different from the prior contrastive learning works. To be specific, we propose a significantly more effective way to incorporate the domain-specific principle of “a molecule can be viewed as a set of meaningful fragments” in molecular contrastive learning frameworks. This unique motivation allows us to consider the relationship between a molecule and its meaningful fragments within a single framework, which differentiates our work from the previous fragment-based molecular representation learning methods that separately consider fragment-level information. Furthermore, HoliMol considers 2D-3D fragment-wise interactions as well, e.g., between 2D fragments and 3D fragments, which allows to learn more fine-grained information of multi-view molecules. This is quite different from prior 2D and 3D contrastive learning methods.
>
> ------
> **[W2] Distinct training configurations (50k, 310k, and 110k) in Table 1,2, and 3.**
>
> We appreciate your comment to clarify this point. We do not create bespoke models for individual tasks, but we carefully design the evaluation setup to validate the applicability of our framework in various real-world scenarios. We first remark that the molecule space is very large, and so the fine-tuning distribution may often be quite different from the pre-training distribution. Therefore, we divide the evaluation setup into two cases based on the pre-training and fine-tuning distributions.
>
> - In Table 1 and Table 2, we assume the pre-training and fine-tuning distributions are different. Specifically, we pre-train on the GEOM dataset and then fine-tune on the MoleculeNet (Table 1) and QM9 (Table 2), respectively.
> - In Table 3, we assume the pre-training and fine-tuning distributions are the same. Specifically, we use the QM9 dataset for both pre-training and fine-tuning.
>
> We also note that we use 50k molecules from the GEOM dataset in Table 1 for a fair comparison with existing works reported in [1]. However, we find that there is no specific reason not to use the full GEOM dataset (310k molecules) for pre-training. Thus, in the evaluation on QM9 (Table 2), we use the entire 310k molecules of the GEOM dataset for pre-training the models.
>
>
> [1] Liu et al., Pre-training Molecular Graph Representation with 3D Geometry, ICLR 2022
>
> -----
> **[W3] Proposed method can be applied to tasks requiring 3D inputs?**
>
>
> Our method can be applied to tasks requiring 3D inputs. Although we mainly focus on fine-tuning on 2D graphs in our main manuscript, we have also shown the effectiveness of our pre-trained 3D GNN when the fine-tuning dataset provides 3D conformation in Table 13 of our original manuscript.
>
>
> -----
> **[W4] Results on the PCQM4Mv2 dataset.**
>
>
> We strongly believe that we have shown the effectiveness of our representation learning method in the common datasets (GEOM and QM9 [1,2]) with various evaluation setups (transfer learning and semi-supervised learning). Indeed, PCQM4Mv2 is mainly used to evaluate molecular architectures or supervised learning methods [3,4]. Thus, we have not considered this dataset since none of the molecular representation learning baselines in our paper conducted experiments on this dataset. Nevertheless, for your information, we have conducted an experiment to verify the effectiveness of our method in the PCQM4Mv2 dataset. Specifically, we fine-tune the pre-trained models of each baseline with a small fraction of the training data, i.e., 10\%, to follow the conventional evaluation setup of molecular representation learning where the labeled data is fewer than unlabeled data. In the table below, our method achieves superior performance in terms of validation MAE. We added the results in Appendix K of the revised manuscript.
>
> \begin{array}{l|ccc|c}
> \hline
> \text{Methods} & \text{-}& \text{3D-Infomax} & \text{GraphMVP-G} & \text{Holi-Mol (Ours)} \newline
> \hline
> \text{Validation MAE} & 0.1703 & 0.1692 & 0.1693 & \textbf{0.1688} \newline
> \hline
> \end{array}
>
>
> [1] Stark et al., 3D Infomax improves GNNs for Molecular Property Prediction, ICML 2022\
> [2] Liu et al., Pre-training Molecular Graph Representation with 3D Geometry, ICLR 2022\
> [3] Liu et al., Gem-2: Next Generation Molecular Property Prediction Network with Many-body and Full-range Interaction Modeling, arXiv 2022\
> [4] Hussain et al., Global Self-Attention as a Replacement for Graph Convolution, KDD 2022

---

> > ### Comment · Reviewer_iXqL · 2024-03-12
> > **Thank you for the response.**
> >
> > W2: I am still quite confused about these settings. Can we just pretrain one model for all these downstream tasks? You can do the same thing for the baselines.
> >
> > W5: Can you also add Transformer-M as a baseline?

---

> > > ### Author Response · Authors · 2024-03-15
> > > **Response to Reviewer iXqL**
> > >
> > > Dear reviewer iXqL,
> > >
> > > We sincerely appreciate your additional comment. We respond to each comment in the following content.
> > >
> > >
> > > -----
> > > **[W2] Evaluation setup.**
> > >
> > > As you point out, it is possible to pre-train one model for all the downstream tasks. Although we believe that our main tables, i.e., Table 1,2, and 3, are crucial to verify the effectiveness of our method in various pre-training/fine-tuning distributions (e.g., transfer learning and semi-supervised learning), we further report the results on the downstream tasks using a single pre-trained model. Specifically, we use the pre-trained model based on the 310k molecules from the GEOM dataset (which is the setup of Table 2) for each method. The tables below show that the single pre-trained model also outperforms the baselines on the downstream tasks (MoleculeNet and QM9). We added this discussion in Appendix P of the revised manuscript.
> > >
> > >
> > > \begin{array}{l|cccccccc|c}
> > > \hline
> > > \text{MoleculeNet} & \text{BBBP} & \text{Tox21} & \text{ToxCast} & \text{Sider} & \text{Clintox} & \text{MUV} & \text{HIV} & \text{Bace} & \text{Avg.}\newline
> > > \hline
> > > \text{3D-Infomax} & 68.8\pm{0.2} & 75.0\pm{1.5} & 64.4\pm{0.4} & 61.7\pm{0.2} & 89.7\pm{1.3} & 76.9\pm{0.8} & 75.3\pm{0.5} & 79.8\pm{0.6} & 73.9 \newline
> > > \text{GraphMVP-G} & 69.8\pm{0.8} & 75.5\pm{0.1} & 64.1\pm{0.2} & 61.1\pm{0.7} & 90.1\pm{0.9} & \textbf{77.6}\pm{0.5} & 75.7\pm{0.5} & 80.3\pm{0.7} & 74.3 \newline \hline
> > > \text{Holi-Mol (Ours)} & \textbf{71.2}\pm{0.8} & \textbf{75.7}\pm{0.3} & \textbf{64.9}\pm{0.4} & \textbf{61.7}\pm{0.2} & \textbf{95.3}\pm{0.2} & \textbf{77.6}\pm{1.4} & \textbf{76.5}\pm{0.4} & \textbf{82.4}\pm{0.5} & \textbf{75.7} \newline
> > > \hline
> > > \end{array}
> > >
> > >
> > >
> > >
> > >
> > > \begin{array}{l|cccccccccccc}
> > > \hline
> > > \text{QM9} & \text{ZPVE} \downarrow & \mu \downarrow & \alpha \downarrow & C_v \downarrow & \text{LUMO} \downarrow & \text{HOMO} \downarrow & \varepsilon_{gap} \downarrow & R^2 \downarrow & U_{0} \downarrow & U_{298} \downarrow & H_{298} \downarrow & G_{298} \downarrow \newline
> > > \hline
> > > \text{3D-Infomax} & 27.0 & 0.051 & 0.355 & 0.126 & 63.4 & 55.2 & 103.8 & 2.99 & \textbf{38.8} & 45.6 & 41.0 & 40.8 \newline
> > > \text{GraphMVP-G} & 24.1 & 0.051 & 0.367 & 0.123 & 59.1 & 53.8 & 100.4 & 2.97 & 39.9 & 44.2 & 41.0 & 40.3 \newline \hline
> > > \text{Holi-Mol (Ours)} & \textbf{24.0} & \textbf{0.049} & \textbf{0.353} & \textbf{0.121} & \textbf{57.1} & \textbf{51.8} & \textbf{97.1} & \textbf{2.90} & 39.2 & \textbf{42.9} & \textbf{40.3} & \textbf{40.0} \newline
> > > \hline
> > > \end{array}
> > >
> > > -----
> > > **[W5] Discussion on Transformer-M.**
> > >
> > > We would like to clarify that Transformer-M [1] cannot be directly compared with our method. Specifically, Holi-Mol proposes a new molecular self-supervised learning framework, while Transformer-M designs an architecture for molecule inputs, i.e., they do not propose a new self-supervised learning framework. Nevertheless, we believe that it would be an interesting direction to apply our GNN-based framework (which we followed the setup of the recent work [2]) to other architectures, including Transformer-M.
> > >
> > >
> > > [1] One Transformer Can Understand Both 2D & 3D Molecular Data, ICLR 2023\
> > > [2] Liu et al., Pre-training Molecular Graph Representation with 3D Geometry, ICLR 2022

---

> ### Comment · Action_Editor_A9Ak · 2024-04-01
> **Request for Final Recommendation**
>
> Dear Reviewer iXqL,
>
> I am reaching out to follow up on the final recommendation for the manuscript you were reviewing, as I noticed that the deadline has been exceeded by 10 days. If you could kindly share your decision regarding this work at your earliest convenience, it would be immensely helpful for us to proceed with the next steps.
>
> Thank you very much for your dedication and time.
>
> Best regards,
> AC

---

### Review · Reviewer_JUTC · 2024-02-20

**Summary Of Contributions:**

This paper proposed a new molecule representation learning framework via semi-supervised learning. The core idea is to leverage the fragment decomposition of molecules to have a new representation parameterization. Regarding the training objective, it combines both the contrastive ones and the predictive one (using 3D information), where the contrastive one tries to provide the contrast between representations from both the 2D and 3D parameterizations.

Experiments on benchmark datasets show that the proposed method is able to improve the performance.

**Audience:**

Yes

**Broader Impact Concerns:**

No concern.

**Claims And Evidence:**

No

**Requested Changes:**

The author claimed that the proposed approach is able to capture both the local and global information, however from the experiment it is not obvious to me. The authors only show the numerical comparison in terms of the final accuracy, but how the ability of obtaining local/global information relates to the accuracy is not clear. So it would be good to provide such kind of studies to show the benefit claimed by the paper.

Furthermore, the fragmentation idea in Eq (2) and (3) seems to be yet another GNN that represent the graphs. It is known that GNNs are able to capture some high-order information, so it is not clear in terms of the motivation, why the explicit fragmentation is needed, or if the author can provide some theoretical justification to this (in terms of the representation power, e.g., try to prove that GNN is not able to represent the fragments or something like that).

Several ablations might be needed as well:
- try to see the performance if we just use the GNN (without the fragmentation) to do the single/cross-view and angle prediction
- see if we just use the fragmented GNN and do the pretraining (without the vanilla GNN)
the purpose is to decouple the representation power of GNN v.s. the objective functions used in the paper.

**Strengths And Weaknesses:**

Strength:
- leveraging the fragmentation of Molecule is a reasonable approach to incorporate more prior information into the representation framework.
- the cross-view contrastive learning is also interesting, where the model would hopefully learn the information across different channels.
- Experimental results seem to be strong.

Weakness:
- Overall the representation learning framework and the multi-view idea is not new. The contribution on the fragmentation is also relatively straightforward and incremental, compared to the existing works.
- Some of the claims are not well supported, and some more ablation studies might be needed.

---

> ### Author Response · Authors · 2024-02-29
> **Response to Reviewer JUTC**
>
> Dear reviewer JUTC,
>
> We sincerely appreciate your efforts in reviewing our manuscript. We respond to each comment in the following content. In the revised manuscript, we have marked the revisions with “blue”.
>
> -----
> **[W1] The representation learning framework and the multi-view idea is not new.**
>
> We politely disagree with the reviewer’s opinion. How we utilize the concept of “fragmentation” is completely different from the prior contrastive learning works. To be specific, we propose a significantly more effective way to incorporate the domain-specific principle of “a molecule can be viewed as a set of meaningful fragments” in molecular contrastive learning frameworks. This unique motivation allows us to consider the relationship between a molecule and its meaningful fragments within a single framework, which differentiates our work from the previous fragment-based molecular representation learning methods that separately consider fragment-level information. Furthermore, HoliMol considers 2D-3D fragment-wise interactions as well, e.g., between 2D fragments and 3D fragments, which allows to learn more fine-grained information of multi-view molecules. This is quite different from prior 2D and 3D contrastive learning methods.
>
> -----
> **[C1] Relationship between local/global information and accuracy.**
>
> Thank you for the opportunity to clarify this point. We first emphasize that the global properties (i.e., labels) of a molecule are closely associated with its meaningful local substructures (i.e., fragments). For example, the ratio of 1-labeled molecules in the BACE (0-1 labeled) dataset is 45%. However, the ratio drops to 29% for samples with cyclohexyl group, and the ratio goes up to 60% for samples with fluoro-benzyl group. These variations can be attributed to the hydrophobicity of the cyclohexyl group and the aromaticity or hydrophilicity of the fluoro-benzyl group. However, the current state-of-the-art methods do not take into account the relationship between the local and global information of molecules, showing sub-optimal performance. Thus, our method aims to capture such a relationship in molecular representation learning.
> For empirical support, we provide t-SNE (see Figure 3 of the revised manuscript) of the molecule representations containing the discussed semantic-determining groups (cyclohexyl and fluoro-benzyl) in the GEOM pre-training dataset. Holi-mol shows superior discriminability for such groups, which supports the overall improvements in downstream tasks in Table 1,2, and 3. This validates the efficacy of our novel approach, which captures the interaction between a molecule and its constituent fragments within a unified framework. We added these respects of discussion in Appendix J of the revised manuscript.
>
> ------
> **[C2-C4] Method clarification.**
>
> We appreciate your comments. However, unfortunately, we find that there is a misunderstanding in our methodology. We would like to clarify the point. Specifically, we do not introduce a “fragmented GNN’’. In our framework, we use a single 2D GNN and a single 3D GNN. For example, Eq. (2), both $z_\mathtt{2D}$ and $z^\mathtt{mix}_\mathtt{2D}$ are obtained from the same 2D GNN. For your information, in Figure 1 of our manuscript, the same GNN is shown with the same color. We hope that this clarification helps you to understand our framework better.

---

### Decision · Action_Editor_A9Ak · 2024-04-23

**Recommendation:** Accept with minor revision

**Comment:**

Reviewer JUTC is the only gatekeeper for this paper. After a thorough examination of the paper, the reviews, and the authors' response, I conclude that the remaining concerns raised by Reviewer JUTC are unfounded. The authors have indeed provided compelling evidence that demonstrates the underperformance of a molecule-level loss approach, which corresponds to the use of a standard GNN. I reached out to Reviewer JUTC for confirmation but have not received a reply.

Given the positive attributes of this work, including the innovative simultaneous consideration of both 2D and 3D perspectives and both holistic and fragmented views, the application of torsional angle prediction, and the strong empirical results, I am inclined to recommend acceptance.

**Audience:**

Yes, AI for science is a rapidly growing field. This work enhances molecule representation learning, which contributes broadly to a diverse array of downstream tasks.

**Claims And Evidence:**

The paper introduces an innovative framework that leverages a multi-view fragmentation approach for the pre-training of molecules, employing both holistic and fragmented molecular representations. This method creatively integrates both 2D and 3D perspectives, presenting a novel approach in molecular studies. To investigate the interactions between molecular fragments, the authors have ingeniously developed a torsional angle prediction method, which aids in elucidating the correlations among fragments.

Prior to the response period, some claims, including the necessity of 2D GNNs, were not fully substantiated. However, the authors' response, supported by supplementary experiments, has effectively addressed these concerns.

It is highly recommended that the authors further refine their paper by providing a clearer explanation of the experimental setup and including the supplementary experiments conducted during the response period.